# Repurposing Foundation Model for Generalizable Medical Time Series Classification

**Nan Huang**
University of North Carolina at Charlotte
NC, US
nhuang1@charlotte.edu

**Haishuai Wang**
Zhejiang University
Zhejiang, China
haishuai.wang@zju.edu.cn

**Zihuai He**
Stanford University
Stanford, CA, US
zihuai@stanford.edu

**Marinka Zitnik**
Harvard University
Boston, MA, US
marinka@hms.harvard.edu

**Xiang Zhang**
University of North Carolina at Charlotte
NC, US
xiang.zhang@charlotte.edu

## Abstract

Medical time series (MedTS) classification suffers from poor generalizability in real-world deployment due to inter- and intra-dataset heterogeneity, such as varying numbers of channels, signal lengths, task definitions, and patient characteristics. To address this, we propose FORMED, a novel framework for repurposing a backbone foundation model, pre-trained on generic time series, to enable highly generalizable MedTS classification on unseen datasets. FORMED combines the backbone with a novel classifier comprising two components: (1) task-specific channel embeddings and label queries, dynamically sized to match any number of channels and target classes, and (2) a shared decoding attention layer, jointly trained across datasets to capture medical domain knowledge through task-agnostic feature-query interactions. After repurposing, FORMED achieves seamless adaptation to unseen MedTS datasets through lightweight label query training (0.1% of parameters), eliminating the need for full fine-tuning or architectural redesign. We evaluate FORMED on 5 diverse MedTS datasets, benchmarking against 11 Task-Specific Models (TSM) and 4 Task-Specific Adaptation (TSA) methods. Our results demonstrate FORMED's dominant performance, achieving up to 35% absolute improvement in F1-score (on ADFTD dataset) over specialized baselines. Further analysis reveals consistent generalization across varying channel configurations, time series lengths, and clinical tasks, which are key challenges in real-world deployment. By decoupling domain-invariant representation learning from task-specific adaptation, FORMED establishes a scalable and resource-efficient paradigm for foundation model repurposing in healthcare. This approach prioritizes clinical adaptability over rigid task-centric design, offering a practical pathway for real-world implementation. Code is available at https://github.com/DL4mHealth/FORMED.

## 1 Introduction

Medical time series (MedTS) classification, such as on electroencephalograms (EEG) and electrocardiograms (ECG), is critical for diagnosing a wide spectrum of medical conditions, including Alzheimer's Disease (AD; Jeong (2004)), Parkinson's Disease (PD; Aljalal et al. (2022b;a)), and heart Arrhythmia (Jin et al., 2024b). Despite significant advancements in developing deep learning models for these tasks, their effective generalization across diverse datasets, sometimes even among individ-

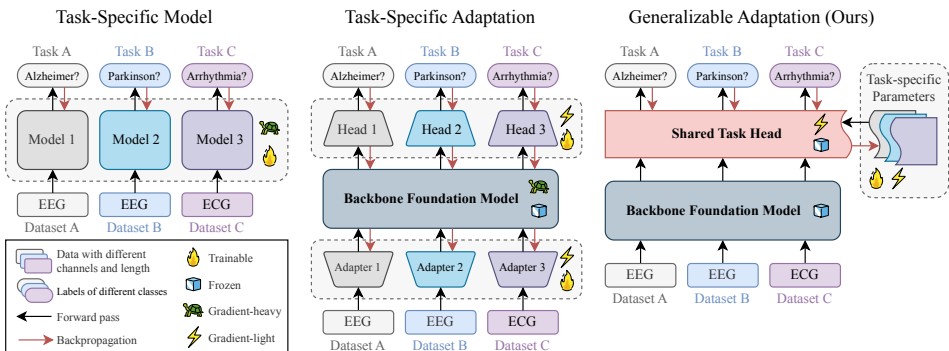

Figure 1: Paradigms of building models for different MedTS classification tasks. **Task-Specific Model (TSM):** Traditional classification models are designed for specific input shape and output classes, thus require retraining from scratch for each new dataset. **Task-Specific Adaptation (TSA):** By using a pre-trained and fixed backbone foundation models, the adaptation to new datasets requires training fewer parameters for each dataset, such as pre- and post-backbone adapters, which makes the combined model no longer applicable to other tasks, lacking generalization across tasks, and more prone to overfitting. **Generalizable Adaptation (GA):** Generalizable adaptation is a post-backbone adaptation module that is shared across tasks of different datasets, which carries domain knowledge and transferable to unseen datasets with training of lightweight task-specific parameters, offering both generalizability and robustness against overfitting.

ual patients within the same dataset, remains a significant hurdle. This limitation critically obstructs the translation of advanced predictive algorithms into reliable real-world clinical applications.

Several unique challenges inherent to MedTS data compound the poor generalizability. Firstly, **inter-dataset heterogeneity** arises from variations in physiological data domains, data collection equipment, and experimental protocols, leading to differences in the *number of channels*, *sample durations*, *sampling rates*, and *diagnostic targets* (Ganapathy et al., 2018). Secondly, **intra-dataset heterogeneity** persists even within single datasets, with variations occurring across recording times, experimental sessions, and, most importantly, *among patients due to unique physiological characteristics* (Wang et al., 2024b; Ganapathy et al., 2018). This often leads to models overfitting to training data and failing to generalize to unseen patients. Thirdly, **data insufficiency** is a persistent issue: the high cost of data collection and privacy concerns often result in small MedTS datasets (Kaushik et al., 2020), making it difficult to train robust models capable of addressing the aforementioned heterogeneity (Ganapathy et al., 2018).

Previous attempts to tackle these issues, such as employing Task-Specific Adaptation (TSA; see Figure 1) in models like Yang et al. (2023), have shown limited success. These methods may unintentionally focus on extracting features relevant only to the initial training task, thereby failing to generalize effectively to new datasets or different medical conditions, as evidenced by marginal or even negative performance gains through pre-training. While on the other hand, recent advancements in foundation models for time series offer a promising avenue. Despite their *predominant focus on forecasting tasks* (Ye et al., 2024; Wen et al., 2022), they demonstrate the ability to learn generic representations of time series data (Liang et al., 2024), thanks to pre-training on large-scale general time series data. This can be beneficial for MedTS classification tasks as well. Our pilot study indicates that directly adapting these models for MedTS classification is better than TSA models trained from scratch, but still fall short in capturing the intricate patterns necessary for specific diagnostic tasks when compared to established Task-Specific Models (TSMs; see Figure 1). This is primarily due to their lack of ability to capture the task-agnostic domain knowledge, which is crucial for generalization across datasets.

To address these limitations, this paper introduces FORMED (**Fo**undation model **R**epurposed for **Med**ical time series classification), a novel approach designed to repurpose foundation models for MedTS classification. FORMED aims to achieve **Generalizable Adaptation** (GA; see Figure 1). This is achieved by utilizing a pre-trained foundation model as its backbone to capture **generic temporal features**, and integrating a novel classifier design to handle MedTS heterogeneity, by architecturally separating **medical domain knowledge** from **task-specific knowledge**. This allows the model to effectively learn and utilize both task-agnostic and task-specific features, enabling

seamless handling of datasets with arbitrary channel configurations, dynamic time series lengths, and diverse diagnostic targets across multiple tasks. Therefore, FORMED enables the backbone model to effectively leverage the commonalities across datasets, while also being flexible enough to adapt to the unique characteristics of each dataset, thus achieving generalization across datasets and tasks.

To facilitate this research, we adopt the comprehensive MedTS cohort curated by (Wang et al., 2024b) as *repurposing cohort*, which comprises five MedTS datasets (two ECG and three EEG). This collection includes approximately 340,000 samples (90 million time-points) in total. These datasets exhibit diverse channel configurations (ranging from 12 to 33 channels), varied diagnostic tasks (from binary neurological to 5-class cardiovascular classification), and differing dataset sizes. This provides different levels of difficulties (both inter- and intra-dataset) for the model to learn from, and serves as a robust training and evaluation platform for the proposed method.

FORMED is strongly supported by empirical results on two aspects: First, for datasets partially included in the repurposing cohort, FORMED achieves state-of-the-art level performance on unseen patients, outperforming 15 TSA and TSM models across all datasets. Second, for completely new datasets not included in the cohort, FORMED can be efficiently adapted by updating only a small proportion of parameters while outperforming the baseline TSA model, and shows robust adapt-time scaling performance with the amount of trainable parameters. This demonstrates the model's ability to generalize across datasets and tasks, and its potential for real-world applications in healthcare.

## 2 RELATED WORK

### 2.1 FOUNDATION MODELS FOR GENERAL TIME SERIES

While recent models like MOMENT (Goswami et al., 2024) and UniTS (Gao et al., 2024) incorporate classification objectives, the majority of foundation model remains heavily concentrated on generative forecasting tasks (Liang et al., 2024). Given their success in forecasting, re-purposing these forecasting-oriented models for MedTS classification is a tempting prospect, yet it presents significant theoretical and practical challenges. These models often have major limitations, such as an inherent design for *univariate time series* in a channel-independent fashion (Nie et al., 2022), and requiring *TSAs* that prevent them from being directly applicable to MedTS classification tasks (Cao et al., 2024; Sun et al., 2024; Chang et al., 2023). Given that medical time series are typically multi-variate, a critical aspect of our repurposing framework is to effectively integrate the information from multichannel features extracted by these backbones.

For instance, models like Time-LLM (Jin et al., 2024a), UniTime (Liu et al., 2024) and GPT4TS (Zhou et al., 2023a) use large language models as backbones. Consequently, they naturally handle time series data in a univariate manner, lacking the ability to integrate information across multiple channels crucial for MedTS classification. Moreover, consistent with the findings of Tan et al. (2024), our empirical observations show that LLM-based time series foundation models do not always achieve optimal performance, even on general time series datasets. Similarly, while TimeGPT (Garza et al., 2024) and TimesFM (Das et al., 2024) are pre-trained on large scale time series data, they typically operate under a channel-independence assumption, treating co-evolving multivariate time series data as a collection of independent univariate series. This shares the same limitation for direct application to multichannel MedTS. Our proposed repurposing framework is specifically designed to address this by incorporating mechanisms to accommodate the multichannel nature of MedTS data, allowing for effective integration of information across channels.

UniTS (Gao et al., 2024) stands out as capable of handling multivariate time series data and has been trained on multiple task domains including classification. However, its scale and design often necessitate fine-tuning the entire model or employing prompt learning for optimal performance. This approach is both computationally expensive (see Figure 1) and data-greedy due to the vast number of parameters to tune, rendering it less suitable for often small-scale MedTS datasets.

Despite their efforts and successes, current foundation models require significant adaptations. Thus, **a key challenge, which FORMED directly addresses, is the adaptation of these powerful but often channel-independent or forecasting-focused models to the multichannel classification demands of MedTS**. This is achieved by integrating dedicated architectural components to address the complexities and multichannel nature of MedTS. While all the aforementioned models represent

potential backbones for our repurposing framework, due to resource limitations and the primary goal to validate the efficacy of our proposed framework, we selected the advanced TimesFM (Das et al., 2024) as our backbone model for this study for its outstanding zero-shot forecasting performance.

## 2.2 ADAPTATION OF FOUNDATION MODELS FOR MEDTS CLASSIFICATION

General-purpose foundation models typically require specific techniques to be effectively adapted for downstream tasks. Common approaches include *prompting*, *fine-tuning*, *re-programming*, and we propose **re-purposing** as a novel approach, each with its own advantages and limitations as summarized in Table 1. We

Table 1: Comparison of adaptation techniques of time series foundation models. Column meanings are in Section A.

| Adaptation | Data Efficiency | New Task Type | Generalizability |
|---|---|---|---|
| Prompting | ✓ | | ✓[1] |
| Fine-tuning | | | ✓ |
| Re-programming | | ✓ | |
| Re-purposing | ✓ | ✓ | ✓ |

will focus here on the distinction between re-programming and re-purposing, as this differentiation is central to our proposed approach for MedTS classification. Further discussion are in Section A.

*Re-programming* often involves reusing a pre-trained model's backbone (*e.g.*, its Transformer layers) without altering its internal weights (Jin et al., 2024a; Chang et al., 2023; Sun et al., 2024; Zhou et al., 2023a), but wrapping it with new input adapters and task-specific output heads (as illustrated by the TSA approach in Figure 1). While this can adapt a model to a new data domain or task type, a significant drawback is that the resulting model often **loses its general-purpose nature**. Both the input adapters and task heads become highly specialized, and **cannot be reused** in future datasets with different configurations, hindering the model's ability to generalize across different tasks, datasets, or to handle the inherent heterogeneity within MedTS (Tan et al., 2024).

*Re-purposing*, as introduced in this work with our FORMED framework, takes a different philosophical approach. It aims to adapt a pre-trained foundation model (often one excelling in tasks like forecasting) to a new class of tasks—in our case, MedTS classification. This is achieved with thoughtful modifications, particularly in how task-specific knowledge is integrated, while striving to maintain the model's core learned representations. The goal is for the repurposed model to serve as a robust and generalizable tool within the target domain (MedTS classification), capable of being efficiently applied to new datasets and diagnostic challenges. Its emphasis on generalizability, data-efficiency, and leveraging domain insights makes re-purposing particularly suitable for adapting powerful, channel-independent time series foundation models to the complexities of multichannel MedTS classification, and creating a new foundation model for the field.

## 2.3 FORECASTING VERSUS CLASSIFICATION IN TIME SERIES

The fundamental differences between time series forecasting and classification are key to understanding the challenges in adapting existing foundation models. **Forecasting** typically involves predicting future sequence values within the same domain as the input, *e.g.*, a sequence → sequence mapping (Lim & Zohren, 2021; Wang et al., 2024a), often by extrapolating patterns **within individual channels**. In contrast, **classification** maps an input sequence to a distinct categorical label, *e.g.*, a sequence → category mapping (Ali et al., 2019). This frequently requires synthesizing complex patterns across **multiple interacting channels** to derive a diagnostic outcome, *e.g.*, diagnosing disease from multichannel EEG (Wan et al., 2023) — a process inherently different from predicting future signal values. This intrinsic divergence in objectives and the nature of data interpretation means that adapting a forecasting foundation model for MedTS classification is **NOT a simple modification of the output layer**. It demands a more comprehensive re-purposing strategy, such as our FORMED framework, designed to bridge these task-specific requirements and effectively handle the complexities of multi-variate medical data.

## 3 PROBLEM STATEMENT

Foundation models, pre-trained on diverse forecasting tasks, have demonstrated a strong capability to capture general time series patterns. Medical waveform data (*e.g.*, EEG, ECG) shares the continuous,

---

[1]Although the model structure is fixed and still applicable to other datasets and tasks, the engineered or learned prompts can be task-specific.

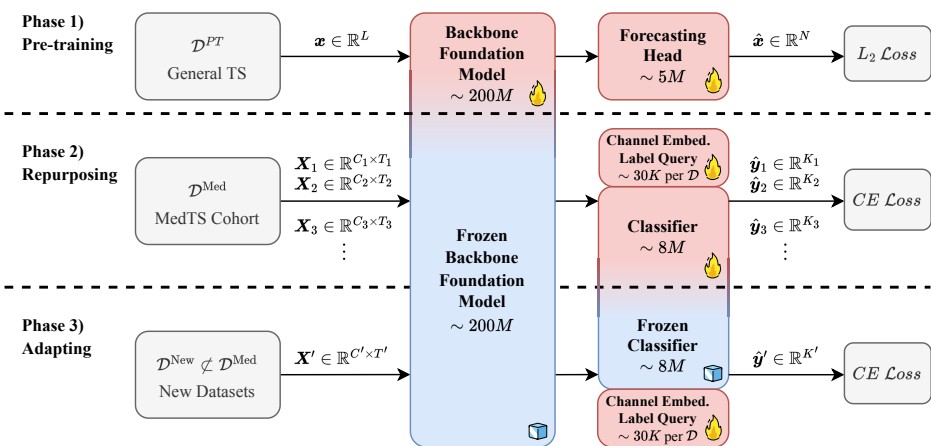

Figure 2: The three-stage process of adapting a time series foundation model for MedTS classification tasks. 1) **Pre-training** is already done on diverse general time series datasets with forecasting tasks. 2) **Repurposing** the foundation model involves changing the forecasting head to a classification head, while keeping the rest of the model fixed, and the new model is then trained on a cohort of MedTS datasets to capture domain knowledge in MedTS. 3) **Adapting** the repurposed model to the new MedTS datasets, where only the minimal task-specific parameters are trained, leveraging the previously learned domain knowledge from the repurposed model.

high-frequency characteristics of the general time series used in pre-training. While sparse, irregular Electronic Health Records (EHR) are also "medical time series," they require fundamentally different tokenization strategies outside the scope of this waveform-centric study. Even with waveform data, transforming pre-trained general time series foundation models into general-purpose classification models, especially for MedTS, is not trivial. This section formally defines the problem and the core concepts underpinning our proposed two-stage adaptation process: repurposing and adapting.

**Definition 3.1. Repurposing**: The process of changing the objective of a pre-trained foundation model to a new class of tasks for which it was not originally trained. This involves introducing and training a relatively small, adaptable output network while keeping the pre-trained backbone fixed.

Let the original pre-trained model consists of a backbone $f : \mathbb{R}^T \to \mathbb{R}^{L \times D}$ for extracting features from a univariate time series of length $T$ into $L$ patched tokens of dimension $D$, and an original task head (*e.g.*, for forecasting, $g : \mathbb{R}^{L \times D} \to \mathbb{R}^N$). We leverage the frozen backbone $f$ for representation learning. For multivariate MedTS input $\boldsymbol{X} \in \mathbb{R}^{C \times T}$ with $C$ channels, the backbone is applied channel-wise to extract features:

$$\mathbf{f} : \mathbb{R}^{C \times T} \to \mathbb{R}^{C \times L \times D} \Leftrightarrow \mathbf{f}(\boldsymbol{X}) := [f(\boldsymbol{X}_{c,:})]_{c=1}^C \tag{1}$$

We then introduce a novel, trainable classification head $h_\theta$. This head is designed to be adaptable to specific task characteristics, such as the number of input channels $C$ and the number of output classes $K$, through learnable task-specific parameters: **Channel Embedding** $\boldsymbol{E} \in \mathbb{R}^{C \times D}$ and **Label Queries** $\boldsymbol{Q} \in \mathbb{R}^{K \times D}$. The mapping becomes:

$$h_\theta|_{\boldsymbol{Q},\boldsymbol{E}} : \mathbb{R}^{C \times L \times D} \to \Delta^K \Rightarrow (h_\theta \circ \mathbf{f})|_{\boldsymbol{Q},\boldsymbol{E}} : \quad \mathbb{R}^{C \times T} \to \Delta^K \tag{2}$$

where $\Delta^K = \left\{ \boldsymbol{d} \in [0,1]^K : \sum_{i=1}^K d_i = 1 \right\}$ is the probability simplex for $K$ classes.

During the **repurposing stage**, $h_\theta$ containing shared parameter $\theta$ along with collections of task-specific embeddings $\mathbf{E} = \{\boldsymbol{E}_i\}$ and $\mathbf{Q} = \{\boldsymbol{Q}_i\}$ are trained across a cohort of diverse MedTS datasets $\mathcal{D}^{\mathrm{Med}} = \left\{ \mathcal{D}_i^{\mathrm{Med}} \right\}$. The objective is to learn domain knowledge for MedTS classification within $\theta$. This translates to minimizing the cross-entropy loss $\mathcal{L}_{\mathrm{CE}}$:

$$\theta^*, \mathbf{E}^*, \mathbf{Q}^* = \underset{\theta, \mathbf{E}, \mathbf{Q}}{\arg\min} \, \mathbb{E}_{i,(\boldsymbol{X}_i, \boldsymbol{y}_i) \in \mathcal{D}_i^{\mathrm{Med}}} \left[ \mathcal{L}_{\mathrm{CE}} \left( h_\theta|_{\boldsymbol{Q}_i, \boldsymbol{E}_i} \left( \mathbf{f}(\boldsymbol{X}_i) \right), \boldsymbol{y}_i \right) \right] \tag{3}$$

**Definition 3.2. Adapting**: The process of applying the repurposed model (with fixed $\theta^*$ and frozen $\mathbf{f}$) to new, unseen MedTS datasets or tasks. This involves learning only a minimal set of new task-specific parameters (new $\boldsymbol{E}'$ and $\boldsymbol{Q}'$) for the new dataset.

For a new dataset $\mathcal{D}^{\text{New}}$ (with potentially different $C'$ channels, $T'$ time steps, and $K'$ classes), the pre-trained backbone $\mathfrak{f}$ and the shared parameters $\theta^*$ of the classifier $h_{\theta^*}$ remain frozen. Only new Channel Embeddings $\boldsymbol{E}' \in \mathbb{R}^{C' \times D}$ and Label Queries $\boldsymbol{Q}' \in \mathbb{R}^{K' \times D}$ are initialized and trained:

$$\boldsymbol{E}'^*, \boldsymbol{Q}'^* = \arg\min_{\boldsymbol{E}', \boldsymbol{Q}'} \mathbb{E}_{(\boldsymbol{X}', \boldsymbol{y}') \in \mathcal{D}^{\text{New}}} \left[ \mathcal{L} \left( h_{\theta^*}|_{\boldsymbol{Q}', \boldsymbol{E}'} \left( \mathfrak{f}(\boldsymbol{X}') \right), \boldsymbol{y}' \right) \right] \tag{4}$$

This two-stage process (illustrated in Figure 2) allows the model to learn general MedTS domain knowledge and efficiently specialize to new tasks with minimal new data and computation.

## 4 MODEL ARCHITECTURE

Our FORMED framework repurposes a pre-trained time series foundation model to serve as a versatile MedTS classification tool. It comprises two main parts: a frozen backbone feature extractor and our novel attention-based classifier designed for adaptability and generalizability.

### 4.1 BACKBONE FEATURE EXTRACTOR FOR VARIABLE LENGTH TIME SERIES

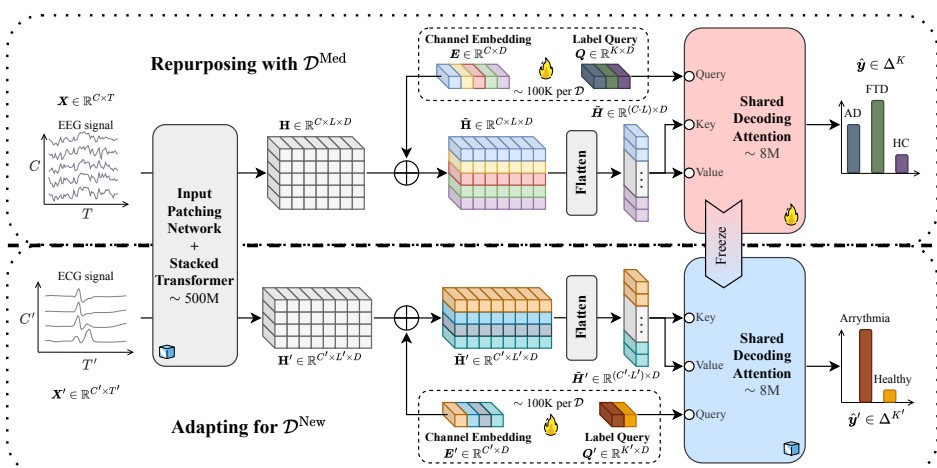

Figure 3: The architecture of the proposed model in repurposing and adapting. The backbone foundation model acts as a feature extractor and remains frozen all the time. The **Channel Embeddings** (CEs) and **Label Queries** (LQs) are task-specific parameters that are learned during both repurposing and adapting, and new ones will be created and learned if encountering new datasets. The **Shared Decoding Attention** (SDA) is a shared Transformer decoder layer that captures the interaction between all the features and classes, which once get trained on curated MedTS datasets $\mathcal{D}^{\text{Med}}$ during repurposing, will be fixed and reused when adapting to all future datasets and tasks $\mathcal{D}^{\text{New}}$. The $\oplus$ denotes broadcast addition.

We employ a powerful pre-trained time series foundation model as the backbone feature extractor. For this work, we selected TimesFM (Das et al., 2024) due to its demonstrated ability to capture complex temporal patterns from variable-length time series, pre-trained on a vast and diverse corpus of general time series data. The backbone's primary role is to transform an input univariate time series $\boldsymbol{x} \in \mathbb{R}^T$ into a sequence of rich feature tokens $\mathbf{H} \in \mathbb{R}^{L \times D}$. As defined in Equation (1), for multichannel MedTS data $\boldsymbol{X} \in \mathbb{R}^{C \times T}$, this backbone is applied independently to each channel, yielding stacked feature tokens $\mathbf{H} \in \mathbb{R}^{C \times L \times D}$. The internal architecture of TimesFM (e.g., its patching mechanism and Transformer layers) is kept frozen throughout our framework and detailed in Section B. The crucial output for our purpose is $\mathbf{H}$, which serves as the input to our novel classifier.

### 4.2 ATTENTION-BASED CLASSIFIER FOR VARYING CHANNEL AND CLASS

Traditional approaches often use a simple linear layer or convolution layer for classification (Zerveas et al., 2021; Yang et al., 2023), which can be restrictive when dealing with the inherent variability of MedTS data (e.g., varying numbers of channels, classes, and sequence lengths). To address these unique challenges, we propose a novel attention-based classifier built upon a Transformer decoder

layer (Vaswani et al., 2017), inspired by advancements in vision tasks (Carion et al., 2020; Meng et al., 2023) but with key modifications tailored for MedTS. This classifier comprises three main components: **C**hannel **E**mbeddings, **L**abel **Q**ueries, and a **S**hared **D**ecoding **A**ttention mechanism.

**Channel Embeddings (CEs)**. MedTS are inherently multi-variate and heterogeneous, often varying in channel configuration across datasets. To address this specificity of medical data without altering the backbone architecture, we introduce learnable Channel Embeddings $\boldsymbol{E} \in \mathbb{R}^{C \times D}$. This allows the model to decouple the spatial topology of the specific medical modality from the generalizable temporal features. For a given dataset with $C$ channels, these embeddings are broadcast-added to the corresponding channel-wise feature tokens $\mathbf{H}$ (from Equation (1)) to produce "channel-aware" feature tokens $\tilde{\mathbf{H}} \in \mathbb{R}^{C \times L \times D}$:

$$\tilde{\mathbf{H}}_{c,l,:} = \mathbf{H}_{c,l,:} \oplus \boldsymbol{E}_{c,:} \quad \forall l \in \{1, 2, \cdots, L\}, c \in \{1, 2, \cdots, C\} \tag{5}$$

These CEs are task-specific and learned during both repurposing (part of $\mathbf{E}$ in Equation (3)) and adapting ($\boldsymbol{E}'$ in Equation (4)).

**Label Queries (LQs)**. To handle varying numbers of diagnostic classes ($K$) per task and provide distinct learnable "anchors" for each class, we use Label Queries $\boldsymbol{Q} \in \mathbb{R}^{K \times D}$. Each row $\boldsymbol{Q}_{i,:}$ is a learnable embedding representing the $i$-th class. These queries actively seek evidence for their respective classes within the channel-aware feature tokens. Like CEs, LQs are task-specific and learned during repurposing and adapting. In practice, we employ $k$ learnable queries for each class to capture potentially complex or multiple defining patterns, where $k$ is a hyperparameter determining the number of distinct "perspectives" or "sub-pattern detectors" for each class. This results in a total of $K \cdot k$ queries in $\boldsymbol{Q} \in \mathbb{R}^{(K \cdot k) \times D}$. Each group of $k$ queries corresponding to a specific class independently attends to the feature tokens to gather evidence.

**Shared Decoding Attention (SDA)**. The core of our classifier is the SDA mechanism, a single Transformer decoder layer whose parameters ($\theta$ in Equation (3)) are shared across all datasets in the repurposing cohort and then frozen during adaptation to new tasks. The SDA takes all $K \cdot k$ Label Queries from $\boldsymbol{Q}$ as queries, and the flattened, channel-aware feature tokens $\texttt{Flatten}(\tilde{\mathbf{H}}) \in \mathbb{R}^{(C \cdot L) \times D}$ as keys and values. It performs multi-head attention followed by an $\texttt{FeedForwardNetwork}$ to produce an initial set of logits $\hat{\boldsymbol{y}}_{\text{raw}} \in \mathbb{R}^{K \cdot k}$:

$$\hat{\boldsymbol{y}}_{\text{raw}} = \texttt{FeedForwardNetwork} \left( \texttt{MultiHeadAttention}(\boldsymbol{Q}, \texttt{Flatten}(\tilde{\mathbf{H}}), \texttt{Flatten}(\tilde{\mathbf{H}})) \right) \tag{6}$$

Since we have $k$ queries (and thus $k$ raw logits) for each of the $K$ classes, these $k$ logits are then averaged to produce a single, final logit for each class, resulting in the final class logits $\hat{\boldsymbol{y}} \in \mathbb{R}^K$:

$$\hat{\boldsymbol{y}}_j = \frac{1}{k} \sum_{i=1}^{k} (\hat{\boldsymbol{y}}_{\text{raw}})_{(j-1)k+i} \quad \forall j \in \{1, 2, \cdots, K\} \tag{7}$$

These final logits $\hat{\boldsymbol{y}}$ are then used with a $\texttt{softmax}$ function for probability prediction and loss calculation. Critically, the parameters within the $\texttt{MultiHeadAttention}$ and the $\texttt{FeedForwardNetwork}$ (collectively $\theta$) are independent of the specific number of input channels $C$, token length $L$, or the total number of queries $K \cdot k$. This architectural choice forces SDA to learn generalizable interaction patterns while allowing for richer, more nuanced class representations.

## 5 EXPERIMENTS

Here we describe the experimental setup, including the datasets chosen to evaluate FORMED against key MedTS challenges, the baselines selected to highlight the advantages of our repurposing framework, and evaluation metrics. Additional training details are included in Section C.

**Datasets**. To thoroughly evaluate FORMED, we use a curated **MedTS cohort** (Wang et al., 2024b) of 5 diverse datasets for the crucial repurposing stage (Figure 2). This cohort, comprising two ECG datasets and three EEG datasets (details in Table 3), provides a breadth of configurations and task types. This enables the **learning of robust, generalizable MedTS domain knowledge** during repurposing. Critically, these datasets span a variety of combinations of channels, sampling rates, sample durations, diagnostic labels, and overall sizes. This inherent diversity is intentional, enabling us to directly **assess FORMED's ability to handle inter-dataset heterogeneity** during repurposing.

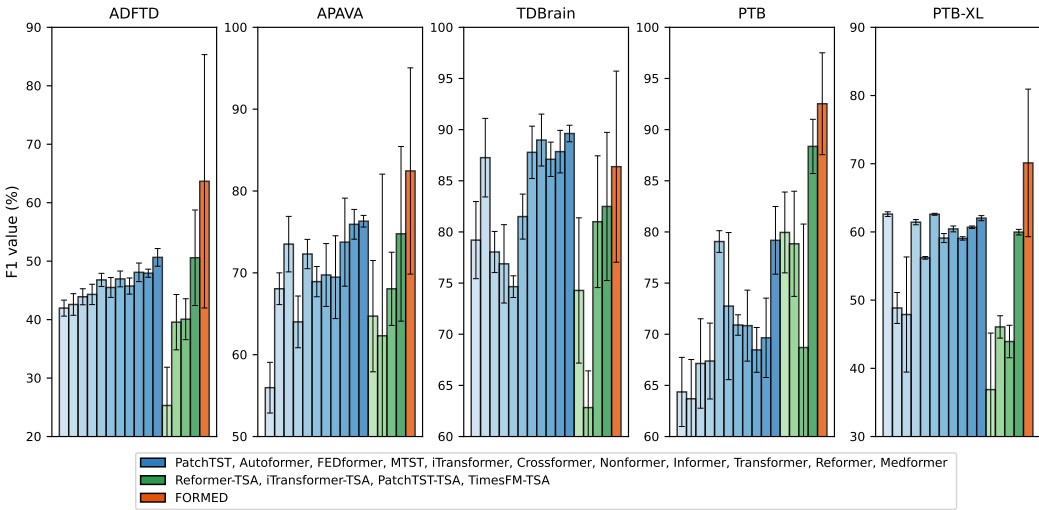

Figure 4: In-domain F1 performance on the MedTS cohort datasets. FORMED achieves SOTA level performance across all datasets in all metrics. Numerical results are shown in Table 4. Other metrics are included in Section E.

Furthermore, all datasets within the cohort and for subsequent adaptation are split following strict **patient-independent** settings (Wang et al., 2024b; Wang et al.). This ensures that the test sets contain subjects entirely unseen during training, rigorously **evaluating the model's robustness against intra-dataset heterogeneity** and its capacity to generate new patients rather than memorizing subject-specific features.

To specifically test the adapting stage and FORMED's generalization to entirely new, unseen tasks and potentially different data characteristics, we also include out-of-domain datasets (ECG200 (Olszewski) and StandWalkJump (Behravan et al.)). Performance on these datasets, especially with **limited data for adaptation**, will demonstrate FORMED's **utility under data insufficiency**.

**Baselines**. We compare FORMED with 15 baselines, including 11 established TSM (Wang et al., 2024b) and 4 TSA models for direct comparison. The TSM models are trained independently on each dataset, including Autoformer (Wu et al., 2021), Crossformer (Zhang & Yan, 2022), FEDformer (Zhou et al., 2022b), Informer (Zhou et al., 2021), iTransformer (Liu et al., 2023), MTST (Zhang et al., 2024), Nonformer (Liu et al., 2022), PatchTST (Nie et al., 2022), Reformer (Kitaev et al., 2020), Transformer (Vaswani et al., 2017) and Medformer (Wang et al., 2024b). They represent the conventional approach, used to verify the benefits of FORMED's repurposing stage in learning transferable domain knowledge.

The TSA models aim to share knowledge across tasks. *TimesFM-TSA* is a key baseline, created by adapting the same TimesFM (Das et al., 2024) backbone with a task-specific CNN head for classification. This allows for a direct evaluation of the additional benefits provided by FORMED's novel classifier design and the two-stage repurposing/adapting strategy over a more straightforward adaptation of the same foundation model. The additional TSA models, *PatchTST-TSA*, *Reformer-TSA*, and *iTransformer-TSA*, are variants of their TSM counterparts. Their backbones are shared across tasks with task-specific heads, and they are configured to have a similar parameter amount to FORMED for demonstrating the superiority of using pre-trained foundation models.

**Evaluations**. The effectiveness of our method is primarily demonstrated through its classification performance on the test sets, using metrics such as accuracy, precision, recall, F1 score, AUROC, and AUPRC. Strong performance on these patient-independent test sets will underscore FORMED's ability to overcome intra-dataset heterogeneity. The generalization ability to unseen tasks, particularly under conditions of potential data insufficiency and inter-dataset heterogeneity, is evaluated through adapting experiments on the small, out-of-domain datasets. These experiments, crucial for assessing the efficacy of the adapting stage where only minimal parameters (CEs and LQs) are learned, are conducted on five random seeds for all models, with results averaged to ensure reliability.

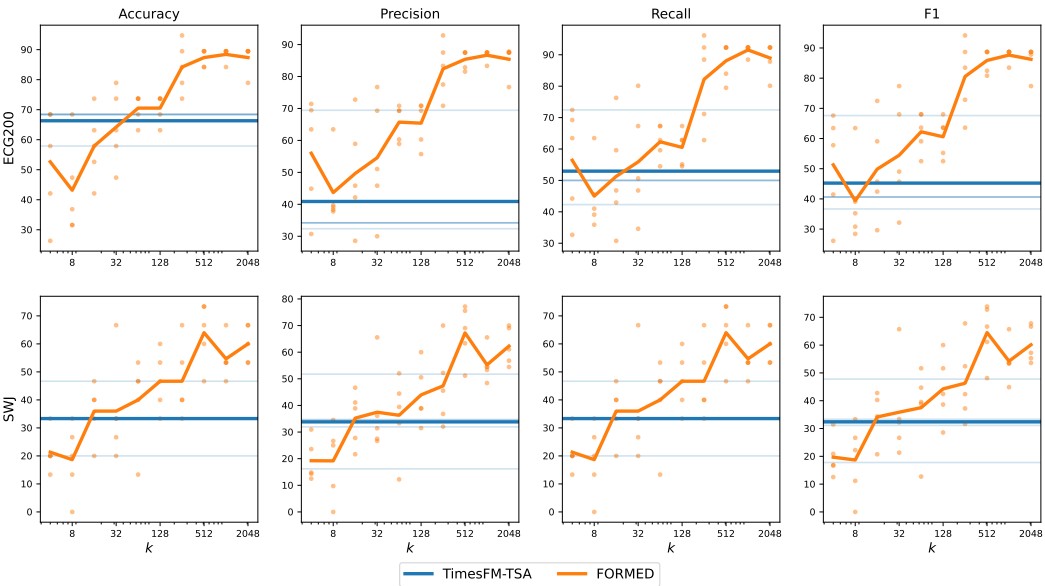

Figure 5: Adapt-time scaling on unseen, out-of-domain dataset. FORMED's performance scales well with $k$ following power law, and outperforms TimesFM-TSA starting from $k = 64$ on ECG200, and from $k = 16$ on StandWalkJump. Numerical results see Table 5.

## 5.1 EVALUATION ON REPURPOSING: GENERALIZE TO UNSEEN SUBJECTS

**Setup**. For repurposing datasets in MedTS cohort, we trained 55 TSM models (11 models for each), and 4 TSA models with 5 task-specific heads each. Our FORMED model is trained on all 5 datasets, using a fixed $k = 16$ for all datasets.

**Results**. The results compellingly demonstrate that the proposed repurposing, which allows the SDA to capture shared MedTS domain knowledge while CEs and LQs handle task-specifics, is more effective than both TSM and TSA approaches for complex classification tasks. As shown in Figure 4, FORMED achieves significant improvements over all baselines. This is particularly pronounced on medium-to-large datasets, such as ADFTD, PTB, and PTB-XL, where performance gains up to 30-40%. On the relatively small TDBrain dataset, which represents a simpler task where TSMs' performance also saturate, FORMED performs on par with the strongest TSMs while maintaining a clear advantage over TSA methods. This superiority stems from FORMED's ability to learn from the collective knowledge of the diverse MedTS cohort, demonstrating that SDA learns generalizable patterns relevant to MedTS classification. Such robust learning across varied datasets translates directly into **addressing intra-dataset heterogeneity**, *i.e.*, better generalization to unseen subjects.

Interestingly, TSA models generally underperform compared to TSMs, and significantly to FORMED. This might suggest that their simpler task-specific heads on a shared backbone are less effective at navigating the substantial inter-dataset heterogeneity within the MedTS cohort than dedicated SDA, CE and LQ architecture. To rigorously verify this, we conducted an extended baseline comparison as shown in Section F, testing simple architectural variants such as replacing our classifier with an MLP or adding LoRA fine-tuning. These modifications still failed to bridge the performance gap (*e.g.*, TimesFM + MLP achieved only 48.84% accuracy on ADFTD *vs.* FORMED's 66.83%), confirming that the foundation model's latent features are not linearly separable and require the non-linear decoding provided by our SDA module.

## 5.2 EVALUATION ON ADAPTING: GENERALIZE TO UNSEEN TASK

**Setup**. In this evaluation, we assess FORMED's ability to generalize to entirely new, unseen tasks, a critical test of its adapting stage and its robustness to **inter-dataset heterogeneity** and **data insufficiency**. We use the TimesFM-TSA model as a strong baseline. The FORMED model is obtained from the repurposing stage; its backbone and SDA parameters are kept frozen. Only newly initialized CEs and LQs are trained for these new tasks, demonstrating **data-efficient adaptation**. We also explore how performance scales by varying the number of queries per class ($k$) from 4 to 2048.

**Results**. The TimesFM-TSA baseline, lacking a sophisticated pre-adaptation to the MedTS domain for its classification components, can struggle to generalize from limited data on new tasks, often overfitting to the training data. In contrast, FORMED, by leveraging the rich MedTS domain knowledge captured in its frozen SDA during the repurposing stage, demonstrates superior adaptation. As seen in Figure 5, FORMED generally outperforms the TimesFM-TSA baseline on these unseen datasets, even with less adaptable parameters (FORMED at $k = 16$ outperforms TimesFM-TSA with only $\sim 1/6$ of the parameter). Notably, the baseline TimesFM-TSA exhibits high variance (*e.g.*, $\pm 12.63\%$ std in F1 on ECG200) due to overfitting. In contrast, FORMED stabilizes significantly as the number of queries $k$ increases. The performance follows a power law improvement; at $k = 1024$, FORMED achieves 87.65% accuracy with reduced variance ($\pm 2.33\%$), demonstrating that the MoE design provides robust density estimation even with limited training samples. This scalability and efficiency in the adapting stage validates FORMED's **strength against data insufficiency and inter-dataset heterogeneity**, making it particularly well-suited for real-world clinical applications where new diagnostic tasks may emerge with **limited available data**.

## 6 DISCUSSIONS AND CONCLUSION

In this paper, we introduced **FORMED**, a novel framework that repurposes general time series foundation models for robust and adaptable MedTS classification. FORMED's core architectural innovation lies in its attention-based classifier, featuring SDA, CEs, and LQs. This design uniquely equips models to **handle variable input lengths, diverse channel configurations, and dynamic numbers of output classes**, addressing limitations of conventional TSM and TSA approaches.

Our comprehensive experiments demonstrate FORMED's strong generalization capabilities, achieving **state-of-the-art performance for unseen patients within datasets (intra-dataset) and effectively adapting to unseen tasks (inter-dataset).** This highlights two significant findings: first, the feasibility and effectiveness of leveraging powerful pre-trained foundation models as backbones for complex MedTS classification. Second, it validates the superiority of our FORMED repurposing framework. The framework excels at capturing transferable MedTS domain knowledge within its shared components during an initial repurposing stage, which then **enables highly efficient and data-scarce adaptation to new tasks by learning only minimal, task-specific parameters.**

Despite these promising results, we acknowledge limitations. Our validation primarily utilized TimesFM as the backbone. While this serves as a strong proof-of-concept for the FORMED framework's efficacy, its performance and interaction with other diverse time series foundation models warrant future investigation. For example, replacing TimesFM with other foundation models that can handle irregularly sampled data or categorical data, will grant FORMED broader applicability. Additionally, the composition and scale of the MedTS cohort employed during the repurposing stage may also influence the breadth of the captured domain knowledge and the overall quality of the learned representations. For instance, we observe that the performance of FORMED during both repurposing (Table 4) and adaptation (Table 5) shows higher variance than other models, which may be attributed to the limited number and scale of datasets in the MedTS cohort and the impact of different splitting by chance. As evidenced in Figure 6, larger datasets like PTB and PTB-XL generally exhibit more stable training trajectories, while smaller datasets like ADFTD and APAVA show more variability across seeds and splits. Future work can explore expanding the pre-training cohort and incorporating advanced joint training strategies to stabilize the FORMED framework.

In conclusion, FORMED presents a significant step towards more **generalizable**, **adaptable**, and **data-efficient** adaptation framework for time series foundation models, as well as a practical deep learning solutions for the unique challenges posed by medical time series analysis.

### ACKNOWLEDGEMENTS

This work is partially supported by the U.S. National Science Foundation under Grant No. 2245894. Any opinions, findings, conclusions or recommendations expressed in this material are those of the authors and do not necessarily reflect the views of the funders.

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

## A  COMPARISON OF ADAPTATION TECHNIQUES IN FOUNDATION MODELS

As discussed in Section 2, adaptation techniques for foundation models mainly includes *Prompting*, *Fine-tuning*, *Re-programming*, and *Re-purposing*. We have introduced re-programming and re-purposing, and here we provide a brief overview of the rest, prompting and fine-tuning, and compare these techniques based on three aspects: *Data Efficiency*, *New Task Type*, and *Generalizability*.

*Prompting & Fine-tuning*: Both are common adaptation techniques for foundation models, where prompting involves conditioning the model with specific instructions or cues, either handcrafted (Zhou et al., 2023b; Reynolds & McDonell, 2021) or learned through data (Zhou et al., 2022a; Liu et al.), and fine-tuning involves updating the model's internal parameters on dedicated dataset (Howard & Ruder, 2018; Ding et al., 2023). While they focus on different aspects of adaptation, they share the commonality of not altering the model's core architecture, therefore the functionality of the model remains unchanged, *e.g.*, model for forecasting remains a forecasting model, and dedicated task head is still required if the task changes (Liu et al.). Moreover, fine-tuning is often more data-intensive, as it necessitates updating the entire model's parameters, whereas prompting typically only requires learning a limited set of task-specific embeddings or prompts.

In general, these techniques can be categorized based on three aspects: *Data efficiency*, as the scale of dataset used for adaptation, typically closely related to and thus constrained by the number of parameters updated; *New Task Type*, as the ability to adapt to new tasks that are different from the original task, such as from forecasting to classification; and *Generalizability*, as the ability for the adapted model to be used on unseen datasets and share knowledge across tasks. Table 1 provides a comparison of these techniques based on these aspects.

## B  IMPLEMENTATION DETAILS

We take TimesFM (Das et al., 2024) as the backbone for repurposing based on our preliminary comparative analysis of existing time series foundation models. TimesFM is pre-trained on a largest-scale dataset of diverse time series data for forecasting tasks and is able to capture general time series patterns within dynamic length of historical input. To repurpose it for MedTS classification, we can break down the model's anatomy into three parts, the input patching network, the stacked Transformer, and the output prediction network.

**Input Patching Network.** Given a univariate time series input $\boldsymbol{x} \in \mathbb{R}^T$ and binary mask $\boldsymbol{m} \in \{0,1\}^T$ with length $T$, they are first broken up into patches $\boldsymbol{X} \in \mathbb{R}^{L \times P}$ and $\boldsymbol{M} \in \{0,1\}^{L \times P}$ in a non-overlapping fashion, where $P$ is the patch size and $L = \lceil \frac{T}{P} \rceil$ is the number of tokens. Each patch $\boldsymbol{X}_{i,:}$ is the concatenation of $P$ consecutive elements of the input sequence $\boldsymbol{x}$ in a non-overlapping fashion and so is the $\boldsymbol{M}_{i,:}$. The $\boldsymbol{X}_{i,:}$ and $\boldsymbol{M}_{i,:}$ denote the $i$-th row of $\boldsymbol{X}$ and $\boldsymbol{M}$, respectively. The sequence of patches $\boldsymbol{X}$ and $\boldsymbol{M}$ are then projected to a sequence of tokens $\boldsymbol{Z} \in \mathbb{R}^{L \times D}$ in the model dimension $D$ using an input residual block:

$$\boldsymbol{Z}_{i,:} = \texttt{InputResidualBlock}(\boldsymbol{X}_{i,:}; \boldsymbol{M}_{i,:}) \tag{8}$$

**Stacked Transformer.** Before passing into the stacked Transformer, the positional encoding will be added to the tokens to form the input sequence $\tilde{\boldsymbol{Z}} \in \mathbb{R}^{L \times D}$. The stacked Transformer is then applied to the input sequence $\tilde{\boldsymbol{Z}}$ to capture the temporal dependencies and extract features using casual self-attention, outputting feature rich tokens $\boldsymbol{H} \in \mathbb{R}^{L \times D}$:

$$\begin{aligned} \tilde{\boldsymbol{Z}}_{i,:} &= \boldsymbol{Z}_{i,:} \oplus \texttt{PositionalEncoding}(i) \\ \boldsymbol{H}_{i,:} &= \texttt{StackedTransformer}(\tilde{\boldsymbol{Z}}_{1,:}, \tilde{\boldsymbol{Z}}_{2,:}, ..., \tilde{\boldsymbol{Z}}_{i,:}; \dot{m}_1, \dot{m}_2, ..., \dot{m}_i) \end{aligned} \tag{9}$$

where $\dot{m}_i = \min\{\boldsymbol{M}_{i,:}\}$ is the mask for the $i$-th patch for masking out completely empty ones.

**Output Prediction Network.** The output prediction network is a residual block layer that maps the last output $\boldsymbol{H}_{L,:}$ from the Transformer back to the original input spaces $\hat{\boldsymbol{x}} \in \mathbb{R}^N$, forming the prediction of the next $N$ time steps:

$$\hat{\boldsymbol{x}} = \texttt{OutputResidualBlock}(\boldsymbol{H}_{L,:}) \tag{10}$$

In summary, the duty of prediction lies solely on the last output prediction network, while the input patching network plus the stacked Transformer can be viewed as a feature extractor that maps the

input time series $x$ to a sequence of feature tokens $H$ (Figure 3). This can be easily extended to process multivariate MedTS by processing each channel of input individually and stack the extracted features as $H \in \mathbb{R}^{C \times L \times D}$ for data of $C$ channels. This will serve as the backbone feature extractor for the downstream classification model.

# C EXPERIMENT SETUP

## C.1 EXPERIMENTAL SETUP

The experiments are carried out on several hosts. The following list summarizes the hardware and software configurations used in our experiments:

Table 2: **Environment setup**.

| Host No. | CPU | Memory (GB) | GPU |
|---|---|---|---|
| 1 | Intel Core i9-10900X | 32 | NVIDIA RTX A5000 $\times$ 1 |
| 2 | AMD Ryzen Threadripper PRO 3995WX | 512 | NVIDIA RTX A5000 $\times$ 4 |
| 3 | AMD EPYC 7713 | 1024 | NVIDIA RTX A5000 $\times$ 8 |
| 4 | AMD EPYC 7513 | 256 | NVIDIA RTX A6000 $\times$ 8 |

| Software/Package | Version |
|---|---|
| Python3 | 3.13.3 |
| PyTorch | 2.7.0 |
| CUDA | 12.4 – 12.6 |

## C.2 CODE AVAILABILITY

The code is publicly available at https://github.com/DL4mHealth/FORMED.

## C.3 DATASET AVAILABILITY & PREPROCESSING

The preprocessed datasets in the MedTS cohort are obtained from the authors of Wang et al. (2024b). The datasets used for adapting are loaded through sktime API and no additional preprocessing is performed.

## C.4 DATA SPLITTING

For datasets in the MedTS cohort, the datasets are split into training, validation, and test sets in the ratio of 6:2:2 at patient level according to Wang et al. (2024b); Wang et al.. The adapting datasets are not clearly defined in terms of patient, so we perform the best effort to split the datasets into training, validation, and test sets in the ratio of 8:1:1 at recording level. The splitting is seeded and stratified to ensure that the training, validation, and test sets have similar distributions of the target classes. The training set is used for training the model, the validation set is used for early stopping, and the test set is used for evaluating the model performance.

## C.5 MODEL TRAINING

The frozen pre-trained backbone TimesFM model is the v2 version from the official GitHub repository, with 50 layers and model dimension of 1280, and patch size of 32. The classifier in FORMED is initialized with default initialization method, and the embeddings are initialized to normal distribution with $\mu = 0$ and $\sigma^2 = 0.1$. The model is trained with AdamW optimizer with weight decay of $1 \times 10^{-3}$ and a custom log-normal learning rate schedule. For each epoch $t$, the learning rate is calculated as:

$$\mathcal{LN}(t; \mu, \sigma^2, T) = \begin{cases} 0 & t = 0 \\ \frac{T}{t} \cdot \exp\left(-\frac{(\ln t - \ln T - \mu)^2}{2\sigma^2}\right) & \text{otherwise} \end{cases}$$

$$\mathrm{lr}(t; \mu, \sigma^2, T) = \mathrm{lr}_{\min} + \frac{\mathcal{LN}(t; \mu, \sigma^2, T)}{\max_t \mathcal{LN}(t; \mu, \sigma^2, T)} \cdot (\mathrm{lr}_{\max} - \mathrm{lr}_{\min})$$

(11)

where we use $\mu = 1, \sigma^2 = 1, T = 10, \mathtt{lr}_{\min} = 1 \times 10^{-5}$ and $\mathtt{lr}_{\max} = 1 \times 10^{-3}$ in our experiments. This creates a quick warm-up phase followed by a gradual decay of the learning rate for annealing. The Figure 6 shows the typical training trajectories of FORMED during repurposing.

During each epoch of training, the model is trained on 100 batches of data from each dataset sampled at random order. The batch size is tailored for each dataset according to the available samples in each dataset, making the 100 batches of data approximately equal to one effective iteration of the whole training set. Additionally, gradient clipping of $1.0$ is applied to prevent exploding gradients and overshooting in earlier epochs. The model is trained for 100 epochs with early stopping on the validation set based on the average F1-score with patience of 10 epochs. The same training procedure is applied to all TSA and our method.

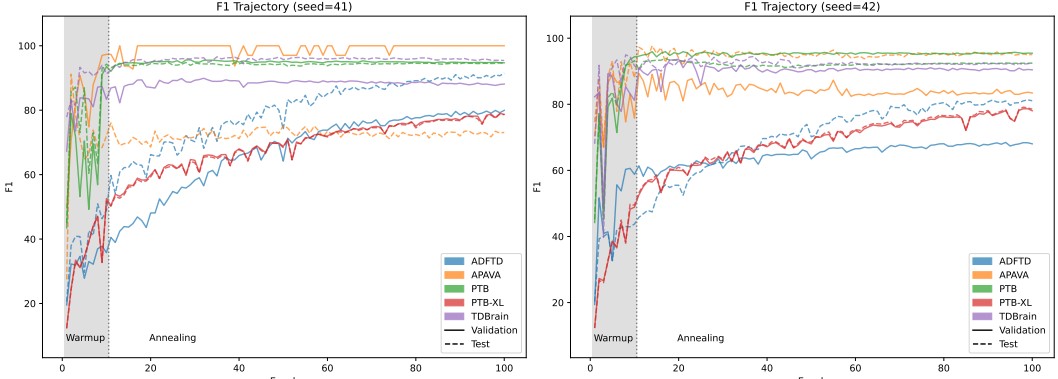

Figure 6: Typical training trajectories of FORMED during repurposing from two different seeds. FORMED can keep improving over a long training period, despite that the gain might not be consistent on different partition of datasets (*e.g.*, validation and test on ADFTD in the right figure), and can vary depending on the exact random seed (*e.g.*, test on APAVA in two figures).

## C.6 HYPERPARAMETER TUNING

The initial hyperparameter $k$ for the classifier in FORMED was not tuned due to the high cost associated with repurposing extra-large models like TimesFM, but rather set to an arbitrary value of $k = 16$ for all datasets. However, we do perform a grid search for the hyperparameter $k$ during the adapting process, as shown in Figure 5. We initiate the search with powers of $4$, until the performance shows no significant improvement. Then we perform a finer search with powers of $2$. We find that $k = 256 \sim 1024$ is a good range for adapting to small datasets. This search is very efficient as the computation cost of increasing $k$ is often negligible compared to the computation cost of large model backbones, despite being linear in $k$ theoretically.

## D  DATASETS

Here we provide the details of the datasets Table 3 used as the MedTS cohort for repurposing in Section 5. The datasets are publicly available, and we follow the pre-processing and splitting procedures as in Wang et al. (2024b).

Table 3: **MedTS Cohort Datasets**.

| Dataset | Type | # Subject | # Sample | Sampling Rate | Sampling Length | # Channel | # Classes |
|---|---|---|---|---|---|---|---|
| ADFTD (Miltiadous et al., 2023; Miltiadous et al.) | EEG | 88 | 69 762 | 256 Hz | 256 | 19 | 3 |
| APAVA (Escudero et al., 2006) | EEG | 23 | 5967 | 256 Hz | 256 | 16 | 2 |
| TDBrain (van Dijk et al., 2022) | EEG | 72 | 6240 | 256 Hz | 256 | 33 | 2 |
| PTB (Goldberger et al., 2000) | ECG | 198 | 64 356 | 250 Hz | 300 | 15 | 2 |
| PTB-XL (Wagner et al., 2020) | ECG | 17 596 | 191 400 | 250 Hz | 250 | 12 | 5 |

# E    EXPERIMENTAL RESULTS

Due to space limitations, here we provide the rest of experimental results of FORMED on the MedTS cohort datasets in terms of accuracy, precision, recall, AUROC, and AUPRC in Figures 7 to 11, along with the full comparison table in Table 4 and adapting results in Table 5.

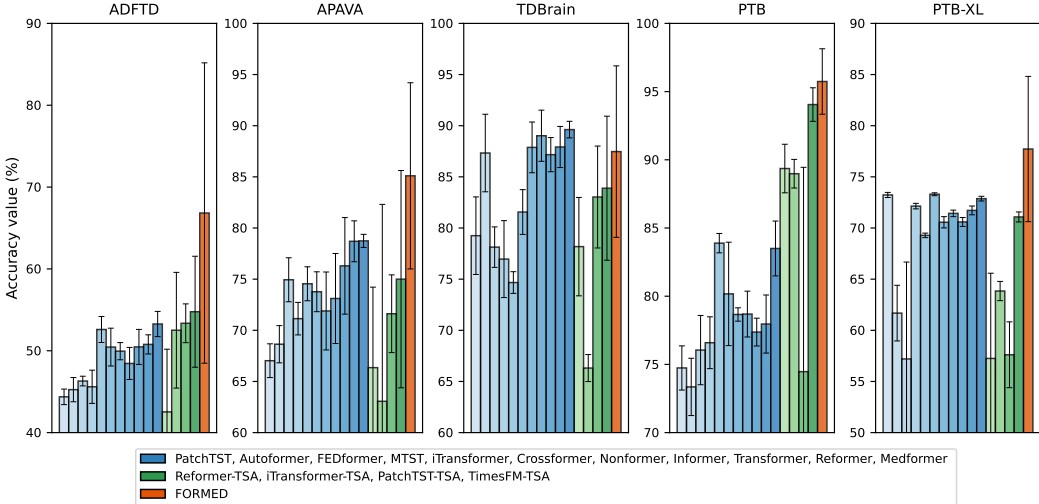

Figure 7: In-domain accuracy performance on the MedTS cohort datasets.

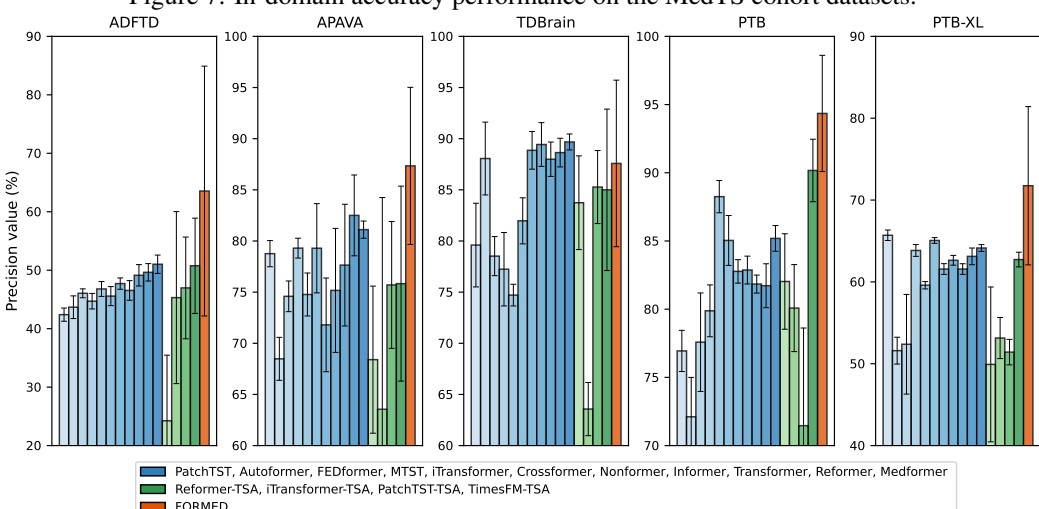

Figure 8: In-domain precision performance on the MedTS cohort datasets.

# F    EXTENDED BASELINE COMPARISON

To rigorously evaluate the effectiveness of FORMED, we conducted an extensive comparison against both traditional machine learning approaches and various configurations of the TimesFM foundation model foundation. The results are summarized in Table 6.

## F.1    BASELINES SETUP

We compared FORMED against MiniRocket (Dempster et al.), a strong machine learning baseline for time series classification. Additionally, we evaluated four distinct variations of the TimesFM architecture to isolate the contributions of our training strategy and classifier design:

- **TimesFM + CNN**: The baseline used in main experiment, *i.e.*, the TimesFM-TSA model, utilizing a CNN head engineered for balanced performance.

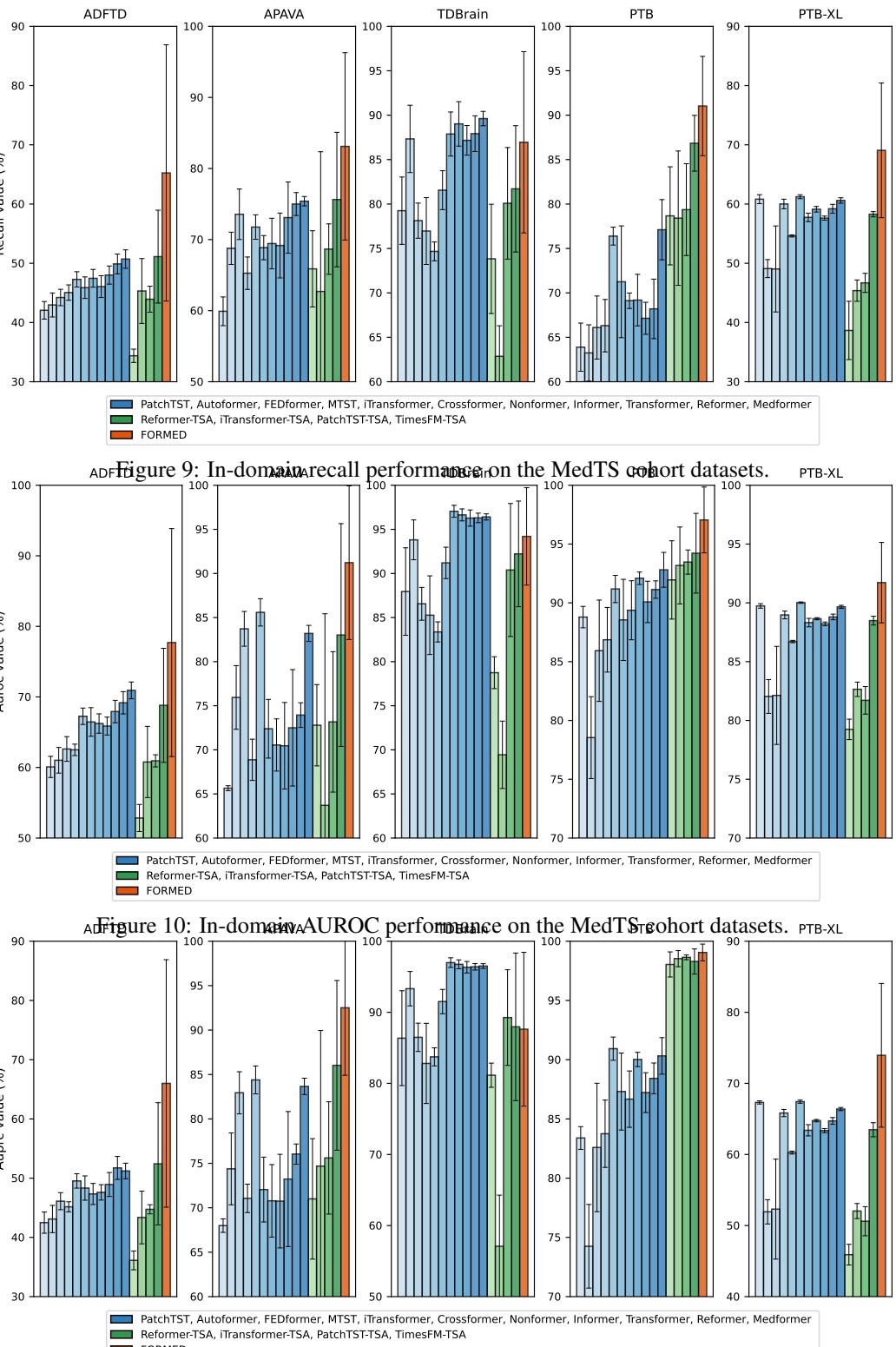

Figure 9: In-domain recall performance on the MedTS cohort datasets.

Figure 10: In-domain AUROC performance on the MedTS cohort datasets.

Figure 11: In-domain AUPRC performance on the MedTS cohort datasets.

- **TimesFM + MLP**: A naive implementation utilizing a linear classifier head (identical to all TSM baselines), serving as a direct probe of the frozen backbone's features.

- **TimesFM + MLP + LoRA**: Extends the MLP baseline by fine-tuning the TimesFM backbone using Low-Rank Adaptation (LoRA, Hu et al.).

- **TimesFM + Attn**: Shares the identical model architecture (Attention-based classifier) as FORMED but is trained individually on each dataset (single-task learning) rather than jointly.

## F.2 PERFORMANCE ANALYSIS

FORMED demonstrates superior performance across the majority of datasets. On the challenging ADFTD dataset, traditional methods like MiniRocket struggle (49.02% accuracy). While the TimesFM + CNN baseline improves this to 54.77%, FORMED significantly outperforms all baselines with an accuracy of 66.83% and an F1 of 63.66%.

Notably, the comparison between TimesFM + Attn and FORMED highlights the efficacy of our joint training paradigm. On PTB, the single-task TimesFM + Attn achieves 93.48% accuracy, whereas the jointly trained FORMED improves this to 95.74%. Similarly, on APAVA, FORMED achieves 85.10% accuracy compared to 65.90% for the single-task equivalent. This validates that the model benefits from the shared representations learned across diverse medical time series tasks.

While simple classifiers like TimesFM + MLP perform competitively on specific, usually smaller datasets like TDBrain (93.60% accuracy), they lack the consistency of FORMED, which maintains high performance across complex multi-class scenarios like PTB-XL, achieving 77.72% accuracy versus the MLP's 65.22%.

For the LoRA-augmented TimesFM + MLP, while it shows improvements over the vanilla MLP on some datasets (*e.g.*, PTB accuracy from 87.12% to 92.76%), it still falls short of FORMED's performance, indicating that merely fine-tuning the backbone is insufficient without the tailored classifier and highly inefficient. Also note that LoRA is orthogonal to classification head design, and can be potentially combined with FORMED for further improvements.

## G  ABLATION STUDY

To justify the architectural components of FORMED, we performed a comprehensive ablation study isolating the effects of Positional Embeddings (PE), Channel Embeddings (CE), and the Mixture of Experts (MoE) strategy (k=16 *vs.* k=1). The results are detailed in Table 7.

The full implementation of FORMED (incorporating PE, CE, and MoE) consistently yields the most robust results. For example, on the ADFTD dataset, removing any single component leads to a significant drop in F1: without PE (46.26%), without CE (42.54%), and without MoE (46.78%), compared to the full model's 63.66%. Similarly, on APAVA, PTB and PTB-XL, the full model outperforms all ablated versions, underscoring the synergistic effect of these components.

While certain ablated versions perform better on specific datasets, such as the configuration without Channel Embeddings achieving marginally higher accuracy on TDBrain (88.08%) compared to the full model (87.47%), the full FORMED model offers the best generalization. It avoids the catastrophic performance drops seen in ablated versions on harder tasks (*e.g.*, the drop to 64.22% on APAVA when CE is removed). This confirms that both PE and CE are essential for handling the structural variety of MedTS, while MoE effectively scales the model's capacity for complex classification boundaries.

Table 4: **Results on MedTS Cohort for disease classification.** Best results from non-TSM models are **bolded** and best results of all models are underlined. TimesFM-TSA shows great improvement over other TSA models and achieves competitive performance with the SOTA TSM models. While our model, FORMED, consistently outperforms the all other model on all datasets.

| Datasets | Adaptation | Models | Accuracy | Precision | Recall | F1 score | AUROC | AUPRC |
|---|---|---|---|---|---|---|---|---|
| **ADFTD** (3-Classes) | TSM | Autoformer | $45.25_{\pm1.48}$ | $43.67_{\pm1.94}$ | $42.96_{\pm2.03}$ | $42.59_{\pm1.85}$ | $61.02_{\pm1.82}$ | $43.10_{\pm2.30}$ |
| | | Crossformer | $50.45_{\pm2.31}$ | $45.57_{\pm1.63}$ | $45.88_{\pm1.82}$ | $45.50_{\pm1.70}$ | $66.45_{\pm2.03}$ | $48.33_{\pm2.05}$ |
| | | FEDformer | $46.30_{\pm0.59}$ | $46.05_{\pm0.76}$ | $44.22_{\pm1.38}$ | $43.91_{\pm1.37}$ | $62.62_{\pm1.75}$ | $46.11_{\pm1.44}$ |
| | | Informer | $48.45_{\pm1.96}$ | $46.54_{\pm1.68}$ | $46.06_{\pm1.84}$ | $45.74_{\pm1.38}$ | $65.87_{\pm1.27}$ | $47.60_{\pm1.30}$ |
| | | iTransformer | $52.60_{\pm1.59}$ | $46.79_{\pm1.27}$ | $47.28_{\pm1.29}$ | $46.79_{\pm1.13}$ | $67.26_{\pm1.16}$ | $49.53_{\pm1.21}$ |
| | | MTST | $45.60_{\pm2.03}$ | $44.70_{\pm1.33}$ | $45.05_{\pm1.30}$ | $44.31_{\pm1.74}$ | $62.50_{\pm0.81}$ | $45.16_{\pm0.85}$ |
| | | Nonformer | $49.95_{\pm1.05}$ | $47.71_{\pm0.97}$ | $47.46_{\pm1.50}$ | $46.96_{\pm1.35}$ | $66.23_{\pm1.37}$ | $47.33_{\pm1.78}$ |
| | | PatchTST | $44.37_{\pm0.95}$ | $42.40_{\pm1.13}$ | $42.06_{\pm1.48}$ | $41.97_{\pm1.37}$ | $60.08_{\pm1.50}$ | $42.49_{\pm1.79}$ |
| | | Reformer | $50.78_{\pm1.17}$ | $49.64_{\pm1.49}$ | $49.89_{\pm1.67}$ | $47.94_{\pm0.69}$ | $69.17_{\pm1.58}$ | $51.73_{\pm1.94}$ |
| | | Transformer | $50.47_{\pm2.14}$ | $49.13_{\pm1.83}$ | $48.01_{\pm1.53}$ | $48.09_{\pm1.59}$ | $67.93_{\pm1.59}$ | $48.93_{\pm2.02}$ |
| | | **Medformer** | $\mathbf{53.27_{\pm1.54}}$ | $\mathbf{51.02_{\pm1.57}}$ | $\mathbf{50.71_{\pm1.55}}$ | $\mathbf{50.65_{\pm1.51}}$ | $\mathbf{70.93_{\pm1.19}}$ | $\mathbf{51.21_{\pm1.32}}$ |
| | TSA | iTransformer-TSA | $52.51_{\pm7.07}$ | $45.31_{\pm14.71}$ | $45.32_{\pm5.47}$ | $39.56_{\pm4.73}$ | $60.78_{\pm5.05}$ | $43.36_{\pm4.45}$ |
| | | PatchTST-TSA | $53.36_{\pm2.37}$ | $46.97_{\pm8.70}$ | $43.93_{\pm2.19}$ | $40.07_{\pm3.50}$ | $60.94_{\pm0.86}$ | $44.74_{\pm0.77}$ |
| | | Reformer-TSA | $42.53_{\pm7.66}$ | $24.24_{\pm11.23}$ | $34.38_{\pm1.12}$ | $25.31_{\pm6.55}$ | $52.84_{\pm1.92}$ | $36.12_{\pm1.59}$ |
| | | TimesFM-TSA | $54.77_{\pm6.78}$ | $50.76_{\pm8.15}$ | $51.12_{\pm7.84}$ | $50.58_{\pm8.17}$ | $68.81_{\pm8.08}$ | $52.42_{\pm10.32}$ |
| | GA | **FORMED (Ours)** | $\underline{66.83_{\pm18.35}}$ | $\underline{63.54_{\pm21.37}}$ | $\underline{65.25_{\pm21.63}}$ | $\underline{63.66_{\pm21.67}}$ | $\underline{77.70_{\pm16.16}}$ | $\underline{66.00_{\pm20.89}}$ |
| **APAVA** (2-Classes) | TSM | Autoformer | $68.64_{\pm1.82}$ | $68.48_{\pm2.10}$ | $68.77_{\pm2.27}$ | $68.06_{\pm1.94}$ | $75.94_{\pm3.61}$ | $74.38_{\pm4.05}$ |
| | | Crossformer | $73.77_{\pm1.95}$ | $79.29_{\pm4.36}$ | $68.86_{\pm1.70}$ | $68.93_{\pm1.85}$ | $72.39_{\pm3.33}$ | $72.05_{\pm3.65}$ |
| | | FEDformer | $74.94_{\pm2.15}$ | $74.59_{\pm1.50}$ | $73.56_{\pm3.55}$ | $73.51_{\pm3.39}$ | $83.72_{\pm1.97}$ | $82.94_{\pm2.37}$ |
| | | Informer | $73.11_{\pm4.40}$ | $75.17_{\pm6.06}$ | $69.17_{\pm4.56}$ | $69.47_{\pm5.06}$ | $70.46_{\pm4.91}$ | $70.75_{\pm5.27}$ |
| | | iTransformer | $74.55_{\pm1.66}$ | $74.77_{\pm2.10}$ | $71.76_{\pm1.72}$ | $72.30_{\pm1.79}$ | $85.59_{\pm1.55}$ | $84.39_{\pm1.57}$ |
| | | MTST | $71.14_{\pm1.59}$ | $79.30_{\pm0.97}$ | $65.27_{\pm2.28}$ | $64.01_{\pm3.16}$ | $68.87_{\pm2.34}$ | $71.06_{\pm1.60}$ |
| | | Nonformer | $71.89_{\pm3.81}$ | $71.80_{\pm4.58}$ | $69.44_{\pm3.56}$ | $69.74_{\pm3.84}$ | $70.55_{\pm2.96}$ | $70.78_{\pm4.08}$ |
| | | PatchTST | $67.03_{\pm1.65}$ | $78.76_{\pm1.28}$ | $59.91_{\pm2.02}$ | $55.97_{\pm3.13}$ | $65.65_{\pm0.28}$ | $67.99_{\pm0.76}$ |
| | | Reformer | $78.70_{\pm2.00}$ | $\mathbf{82.50_{\pm3.95}}$ | $75.00_{\pm1.61}$ | $75.93_{\pm1.82}$ | $73.94_{\pm1.40}$ | $76.04_{\pm1.14}$ |
| | | Transformer | $76.30_{\pm4.72}$ | $77.64_{\pm5.95}$ | $73.09_{\pm5.01}$ | $73.75_{\pm5.38}$ | $72.50_{\pm6.60}$ | $73.23_{\pm7.60}$ |
| | | **Medformer** | $\mathbf{78.74_{\pm0.64}}$ | $81.11_{\pm0.84}$ | $\mathbf{75.40_{\pm0.66}}$ | $\mathbf{76.31_{\pm0.71}}$ | $\mathbf{83.20_{\pm0.91}}$ | $\mathbf{83.66_{\pm0.92}}$ |
| | TSA | iTransformer-TSA | $63.06_{\pm19.25}$ | $63.55_{\pm20.70}$ | $62.70_{\pm19.66}$ | $62.31_{\pm19.75}$ | $63.71_{\pm21.72}$ | $74.69_{\pm15.26}$ |
| | | PatchTST-TSA | $71.62_{\pm3.80}$ | $75.70_{\pm6.20}$ | $68.66_{\pm3.55}$ | $68.05_{\pm4.47}$ | $73.17_{\pm7.96}$ | $75.61_{\pm6.32}$ |
| | | Reformer-TSA | $66.35_{\pm7.87}$ | $68.40_{\pm7.19}$ | $65.88_{\pm5.36}$ | $64.71_{\pm6.80}$ | $72.79_{\pm4.61}$ | $71.00_{\pm6.78}$ |
| | | TimesFM-TSA | $75.00_{\pm10.62}$ | $75.83_{\pm9.53}$ | $75.63_{\pm9.46}$ | $74.77_{\pm10.67}$ | $83.02_{\pm12.63}$ | $86.03_{\pm9.55}$ |
| | GA | **FORMED (Ours)** | $\underline{85.10_{\pm9.10}}$ | $\underline{87.34_{\pm7.68}}$ | $\underline{83.11_{\pm13.18}}$ | $\underline{82.45_{\pm12.6}}$ | $\underline{91.22_{\pm8.71}}$ | $\underline{92.51_{\pm7.60}}$ |
| **TDBrain** (2-Classes) | TSM | Autoformer | $87.33_{\pm3.79}$ | $88.06_{\pm3.56}$ | $87.33_{\pm3.79}$ | $87.26_{\pm3.84}$ | $93.81_{\pm2.26}$ | $93.32_{\pm2.42}$ |
| | | Crossformer | $81.56_{\pm2.19}$ | $81.97_{\pm2.25}$ | $81.56_{\pm2.19}$ | $81.50_{\pm2.20}$ | $91.20_{\pm1.78}$ | $91.51_{\pm1.71}$ |
| | | FEDformer | $78.13_{\pm1.98}$ | $78.52_{\pm1.91}$ | $78.13_{\pm1.98}$ | $78.04_{\pm2.01}$ | $86.56_{\pm1.86}$ | $86.48_{\pm1.99}$ |
| | | Informer | $89.02_{\pm2.50}$ | $89.43_{\pm2.14}$ | $89.02_{\pm2.50}$ | $88.98_{\pm2.54}$ | $96.64_{\pm0.68}$ | $96.75_{\pm0.63}$ |
| | | iTransformer | $74.67_{\pm1.06}$ | $74.71_{\pm1.06}$ | $74.67_{\pm1.06}$ | $74.65_{\pm1.06}$ | $83.37_{\pm1.14}$ | $83.73_{\pm1.27}$ |
| | | MTST | $76.96_{\pm3.76}$ | $77.24_{\pm3.59}$ | $76.96_{\pm3.76}$ | $76.88_{\pm3.83}$ | $85.27_{\pm4.46}$ | $82.81_{\pm5.64}$ |
| | | Nonformer | $87.88_{\pm2.48}$ | $88.86_{\pm1.84}$ | $87.88_{\pm2.48}$ | $87.78_{\pm2.56}$ | $\mathbf{97.05_{\pm0.68}}$ | $\mathbf{96.99_{\pm0.68}}$ |
| | | PatchTST | $79.25_{\pm3.79}$ | $79.60_{\pm4.09}$ | $79.25_{\pm3.79}$ | $79.20_{\pm3.77}$ | $87.95_{\pm4.96}$ | $86.36_{\pm6.67}$ |
| | | Reformer | $87.92_{\pm2.01}$ | $88.64_{\pm1.40}$ | $87.92_{\pm2.01}$ | $87.85_{\pm2.08}$ | $96.30_{\pm0.54}$ | $96.40_{\pm0.45}$ |
| | | Transformer | $87.17_{\pm1.67}$ | $87.99_{\pm1.68}$ | $87.17_{\pm1.67}$ | $87.10_{\pm1.68}$ | $96.28_{\pm0.92}$ | $96.34_{\pm0.81}$ |
| | | **Medformer** | $\mathbf{89.62_{\pm0.81}}$ | $\mathbf{89.68_{\pm0.78}}$ | $\mathbf{89.62_{\pm0.81}}$ | $\mathbf{89.62_{\pm0.81}}$ | $96.41_{\pm0.35}$ | $96.51_{\pm0.33}$ |
| | TSA | iTransformer-TSA | $66.31_{\pm1.32}$ | $63.57_{\pm2.60}$ | $62.86_{\pm3.44}$ | $62.83_{\pm3.59}$ | $69.44_{\pm3.82}$ | $57.08_{\pm7.20}$ |
| | | PatchTST-TSA | $83.03_{\pm4.98}$ | $85.27_{\pm3.57}$ | $80.08_{\pm6.28}$ | $81.00_{\pm6.44}$ | $90.39_{\pm7.54}$ | $89.26_{\pm6.73}$ |
| | | Reformer-TSA | $78.18_{\pm4.81}$ | $83.74_{\pm4.58}$ | $73.83_{\pm6.15}$ | $74.28_{\pm7.10}$ | $78.75_{\pm1.81}$ | $81.15_{\pm1.70}$ |
| | | TimesFM-TSA | $83.89_{\pm7.04}$ | $85.00_{\pm7.89}$ | $81.71_{\pm7.11}$ | $82.49_{\pm7.24}$ | $92.22_{\pm5.99}$ | $\underline{87.96_{\pm10.38}}$ |
| | GA | **FORMED (Ours)** | $\underline{87.47_{\pm8.38}}$ | $87.58_{\pm8.14}$ | $86.95_{\pm10.20}$ | $86.38_{\pm9.34}$ | $94.20_{\pm5.52}$ | $87.62_{\pm10.82}$ |
| **PTB** (2-Classes) | TSM | Autoformer | $73.35_{\pm2.10}$ | $72.11_{\pm2.89}$ | $63.24_{\pm3.17}$ | $63.69_{\pm3.84}$ | $78.54_{\pm3.48}$ | $74.25_{\pm3.53}$ |
| | | Crossformer | $80.17_{\pm3.79}$ | $85.04_{\pm1.83}$ | $71.25_{\pm6.29}$ | $72.75_{\pm7.19}$ | $88.55_{\pm3.45}$ | $87.31_{\pm3.25}$ |
| | | FEDformer | $76.05_{\pm2.54}$ | $77.58_{\pm3.61}$ | $66.10_{\pm3.55}$ | $67.14_{\pm4.37}$ | $85.93_{\pm4.31}$ | $82.59_{\pm5.42}$ |
| | | Informer | $78.69_{\pm1.68}$ | $82.87_{\pm1.02}$ | $69.19_{\pm2.90}$ | $70.84_{\pm3.47}$ | $92.09_{\pm0.53}$ | $90.02_{\pm0.60}$ |
| | | iTransformer | $83.89_{\pm0.71}$ | $\mathbf{88.25_{\pm1.18}}$ | $76.39_{\pm1.01}$ | $79.06_{\pm1.06}$ | $91.18_{\pm1.16}$ | $90.93_{\pm0.98}$ |
| | | MTST | $76.59_{\pm1.90}$ | $79.88_{\pm1.90}$ | $66.31_{\pm2.95}$ | $67.38_{\pm3.71}$ | $86.86_{\pm2.75}$ | $83.75_{\pm2.84}$ |
| | | Nonformer | $78.66_{\pm0.49}$ | $82.77_{\pm0.86}$ | $69.12_{\pm0.87}$ | $70.90_{\pm1.00}$ | $89.37_{\pm2.51}$ | $86.67_{\pm2.38}$ |
| | | PatchTST | $74.74_{\pm1.62}$ | $76.94_{\pm1.51}$ | $63.89_{\pm2.71}$ | $64.36_{\pm3.38}$ | $88.79_{\pm0.91}$ | $83.39_{\pm0.96}$ |
| | | Reformer | $77.96_{\pm2.13}$ | $81.72_{\pm1.61}$ | $68.20_{\pm3.35}$ | $69.65_{\pm3.88}$ | $91.13_{\pm0.74}$ | $88.42_{\pm1.30}$ |
| | | Transformer | $77.37_{\pm1.02}$ | $81.84_{\pm0.66}$ | $67.14_{\pm1.80}$ | $68.47_{\pm2.19}$ | $90.08_{\pm1.76}$ | $87.22_{\pm1.68}$ |
| | | **Medformer** | $\mathbf{83.50_{\pm2.01}}$ | $85.19_{\pm0.94}$ | $\mathbf{77.11_{\pm3.39}}$ | $\mathbf{79.18_{\pm3.31}}$ | $\mathbf{92.81_{\pm1.48}}$ | $\mathbf{90.32_{\pm1.54}}$ |
| | TSA | iTransformer-TSA | $88.98_{\pm1.05}$ | $80.08_{\pm3.19}$ | $78.41_{\pm7.56}$ | $78.84_{\pm5.14}$ | $93.18_{\pm3.27}$ | $98.53_{\pm0.68}$ |
| | | PatchTST-TSA | $74.47_{\pm14.98}$ | $71.46_{\pm7.16}$ | $79.37_{\pm5.17}$ | $68.70_{\pm12.07}$ | $93.47_{\pm1.03}$ | $98.64_{\pm0.20}$ |
| | | Reformer-TSA | $89.36_{\pm1.78}$ | $82.03_{\pm3.50}$ | $78.67_{\pm5.52}$ | $79.95_{\pm3.95}$ | $91.95_{\pm3.33}$ | $98.04_{\pm1.06}$ |
| | | TimesFM-TSA | $94.05_{\pm1.23}$ | $90.17_{\pm2.29}$ | $86.84_{\pm3.14}$ | $88.36_{\pm2.65}$ | $94.22_{\pm3.39}$ | $98.29_{\pm1.06}$ |
| | GA | **FORMED (Ours)** | $\underline{95.74_{\pm2.40}}$ | $\underline{94.35_{\pm4.26}}$ | $\underline{91.03_{\pm5.59}}$ | $\underline{92.53_{\pm4.98}}$ | $\underline{97.05_{\pm2.80}}$ | $\underline{99.05_{\pm0.71}}$ |
| **PTB-XL** (5-Classes) | TSM | Autoformer | $61.68_{\pm2.72}$ | $51.60_{\pm1.64}$ | $49.10_{\pm1.52}$ | $48.85_{\pm2.27}$ | $82.04_{\pm1.44}$ | $51.93_{\pm1.71}$ |
| | | Crossformer | $\mathbf{73.30_{\pm0.14}}$ | $\mathbf{65.06_{\pm0.35}}$ | $\mathbf{61.23_{\pm0.33}}$ | $\mathbf{62.59_{\pm0.14}}$ | $\mathbf{90.02_{\pm0.06}}$ | $\mathbf{67.43_{\pm0.22}}$ |
| | | FEDformer | $57.20_{\pm9.47}$ | $52.38_{\pm6.09}$ | $49.04_{\pm7.26}$ | $47.89_{\pm4.44}$ | $82.13_{\pm4.17}$ | $52.31_{\pm7.03}$ |
| | | Informer | $71.43_{\pm0.32}$ | $62.64_{\pm0.60}$ | $59.12_{\pm0.47}$ | $60.44_{\pm0.43}$ | $88.65_{\pm0.09}$ | $64.76_{\pm0.17}$ |
| | | iTransformer | $69.28_{\pm0.22}$ | $59.59_{\pm0.45}$ | $54.62_{\pm0.18}$ | $56.20_{\pm0.19}$ | $86.71_{\pm0.10}$ | $60.27_{\pm0.21}$ |
| | | MTST | $72.14_{\pm0.27}$ | $63.84_{\pm0.72}$ | $60.01_{\pm0.81}$ | $61.43_{\pm0.38}$ | $88.97_{\pm0.33}$ | $65.83_{\pm0.51}$ |
| | | Nonformer | $70.56_{\pm0.55}$ | $61.57_{\pm0.66}$ | $57.75_{\pm0.72}$ | $59.10_{\pm0.66}$ | $88.32_{\pm0.36}$ | $63.40_{\pm0.79}$ |
| | | PatchTST | $73.23_{\pm0.25}$ | $65.70_{\pm0.64}$ | $60.82_{\pm0.76}$ | $62.61_{\pm0.34}$ | $89.74_{\pm0.19}$ | $67.32_{\pm0.22}$ |
| | | Reformer | $71.72_{\pm0.43}$ | $63.12_{\pm1.02}$ | $59.20_{\pm0.75}$ | $60.69_{\pm0.18}$ | $88.80_{\pm0.24}$ | $64.72_{\pm0.47}$ |
| | | Transformer | $70.59_{\pm0.44}$ | $61.57_{\pm0.65}$ | $57.62_{\pm0.35}$ | $59.05_{\pm0.25}$ | $88.21_{\pm0.16}$ | $63.36_{\pm0.29}$ |
| | | Medformer | $72.87_{\pm0.23}$ | $64.14_{\pm0.42}$ | $60.60_{\pm0.46}$ | $62.02_{\pm0.37}$ | $89.66_{\pm0.13}$ | $66.39_{\pm0.22}$ |
| | TSA | iTransformer-TSA | $63.84_{\pm0.94}$ | $53.14_{\pm2.51}$ | $45.38_{\pm1.80}$ | $46.07_{\pm1.64}$ | $82.65_{\pm0.61}$ | $52.04_{\pm1.06}$ |
| | | PatchTST-TSA | $57.61_{\pm3.22}$ | $51.42_{\pm1.56}$ | $46.69_{\pm1.61}$ | $43.93_{\pm2.38}$ | $81.71_{\pm1.17}$ | $50.61_{\pm2.05}$ |
| | | Reformer-TSA | $57.25_{\pm8.33}$ | $49.92_{\pm9.46}$ | $38.66_{\pm4.92}$ | $36.88_{\pm8.29}$ | $79.24_{\pm0.87}$ | $45.90_{\pm1.46}$ |
| | | TimesFM-TSA | $71.08_{\pm0.49}$ | $62.73_{\pm0.90}$ | $58.28_{\pm0.44}$ | $59.97_{\pm0.41}$ | $88.49_{\pm0.37}$ | $63.48_{\pm0.99}$ |
| | GA | **FORMED (Ours)** | $\underline{77.72_{\pm7.10}}$ | $\underline{71.75_{\pm9.67}}$ | $\underline{69.06_{\pm11.37}}$ | $\underline{70.12_{\pm10.82}}$ | $\underline{91.72_{\pm3.42}}$ | $\underline{73.95_{\pm10.11}}$ |

Table 5: **Results on adapting to unseen datasets.** Best results are highlighted in **bold**.

| Datasets | Model | $k$ factor | Accuracy | Precision | Recall | F1 score | AUROC | AUPRC |
|---|---|---|---|---|---|---|---|---|
| | **TimesFM-TSA** | N/A | 66.32±4.71 | 40.89±15.98 | 52.95±11.39 | 45.23±12.63 | 80.26±9.79 | 91.24±5.22 |
| | | 4 | 52.63±18.23 | 55.99±17.59 | 56.41±17.19 | 51.28±17.23 | 65.90±18.05 | 81.64±10.65 |
| | | 8 | 43.16±15.52 | 43.68±11.08 | 45.00±10.89 | 39.40±14.06 | 45.13±15.84 | 71.92±8.59 |
| | | 16 | 57.89±11.77 | 49.65±16.85 | 51.28±17.36 | 49.86±16.41 | 68.46±20.1 | 85.77±8.72 |
| ECG200 | | 32 | 64.21±12.57 | 54.57±18.70 | 55.90±17.89 | 54.46±18.12 | 63.33±14.72 | 80.85±8.63 |
| | **FORMED** | 64 | 70.53±4.71 | 65.73±5.64 | 62.31±5.45 | 62.24±6.58 | 73.85±7.23 | 88.05±4.42 |
| | | 128 | 70.53±4.71 | 65.39±6.96 | 60.51±5.53 | 60.59±6.47 | 75.90±4.19 | 89.96±1.81 |
| | | 256 | 84.21±8.32 | 82.40±8.57 | 82.18±14.44 | 80.56±12.31 | 89.23±11.13 | 95.35±5.06 |
| | | 512 | 87.37±2.88 | 85.38±2.94 | 88.08±6.01 | 85.87±3.91 | 95.90±4.66 | 98.27±1.97 |
| | | 1024 | **88.42**±**2.35** | **86.67**±**1.86** | **91.54**±**1.72** | **87.65**±**2.33** | **98.21**±**1.15** | **99.23**±**0.50** |
| | | 2048 | 87.37±4.71 | 85.41±4.87 | 88.97±5.31 | 86.25±4.97 | 95.90±3.56 | 98.22±1.56 |
| | **TimesFM-TSA** | N/A | 33.33±9.43 | 33.83±12.62 | 33.33±9.43 | 32.41±10.64 | 52.00±5.79 | 44.89±4.97 |
| | | 4 | 21.33±7.30 | 19.23±7.84 | 21.33±7.30 | 19.69±7.22 | 44.00±10.66 | 42.07±7.44 |
| | | 8 | 18.67±12.82 | 19.18±13.99 | 18.67±12.82 | 18.68±13.19 | 46.13±12.73 | 40.63±7.32 |
| | | 16 | 36.00±10.11 | 35.22±10.23 | 36.00±10.11 | 34.15±8.57 | 53.60±10.18 | 46.11±8.79 |
| StandWalkJump | | 32 | 36.00±18.01 | 37.45±16.15 | 36.00±18.01 | 35.87±17.37 | 57.33±13.64 | 53.11±15.30 |
| | **FORMED** | 64 | 40.00±15.63 | 36.30±15.11 | 40.00±15.63 | 37.50±14.77 | 56.80±14.88 | 50.30±13.12 |
| | | 128 | 46.67±10.54 | 43.96±11.27 | 46.67±10.54 | 44.26±12.08 | 63.60±11.72 | 55.75±13.10 |
| | | 256 | 46.67±13.33 | 47.33±14.87 | 46.67±13.33 | 46.29±14.25 | 63.60±1.80 | 53.81±2.64 |
| | | 512 | **64.00**±**11.16** | **67.25**±**10.53** | **64.00**±**11.16** | **64.52**±**10.48** | **68.13**±**5.06** | **57.27**±**4.25** |
| | | 1024 | 54.67±7.30 | 55.24±6.28 | 54.67±7.30 | 54.23±7.44 | 62.13±1.66 | 53.30±1.42 |
| | | 2048 | 60.00±6.67 | 62.29±7.03 | 60.00±6.67 | 60.12±6.63 | 65.33±2.98 | 56.34±2.27 |

Table 6: **Additional baseline comparison on all datasets.** Best results are **bolded**.

| Datasets | Model | Accuracy | Precision | Recall | F1 score | AUROC | AUPRC |
|---|---|---|---|---|---|---|---|
| | **MiniRocket** | 49.02±0.20 | 41.37±0.19 | 43.64±0.18 | 41.75±0.16 | — | — |
| | **PatchTST + FORMED** | 45.05±2.77 | 45.43±3.91 | 42.04±1.88 | 36.68±1.76 | 59.22±2.06 | 41.10±1.73 |
| ADFTD | **TimesFM + CNN** | 54.77±6.78 | 50.76±8.15 | 51.12±7.84 | 50.58±8.17 | 68.81±8.08 | 52.42±10.32 |
| (3-Classes) | **TimesFM + MLP** | 48.84±2.50 | 41.94±1.43 | 43.47±1.30 | 41.72±1.23 | 60.73±1.40 | 43.06±1.41 |
| | **TimesFM + MLP + LoRA** | 49.23±2.77 | 45.81±3.20 | 46.04±2.29 | 45.41±2.85 | 63.83±2.34 | 46.15±3.08 |
| | **TimesFM + Attn** | 52.77±1.71 | 46.89±1.20 | 46.34±0.53 | 45.25±1.05 | 66.17±0.99 | 48.02±0.81 |
| | **FORMED** | **66.83**±**18.35** | **63.54**±**21.37** | **65.25**±**21.63** | **63.66**±**21.67** | **77.70**±**16.16** | **66.00**±**20.89** |
| | **MiniRocket** | 67.08±4.63 | 71.38±3.51 | 70.17±4.17 | 66.90±4.82 | — | — |
| | **PatchTST + FORMED** | 69.91±9.91 | 69.11±10.86 | 68.54±11.81 | 68.14±11.11 | 79.32±11.33 | 86.6±7.26 |
| APAVA | **TimesFM + CNN** | 75.00±10.62 | 75.83±9.53 | 75.63±9.46 | 74.77±10.67 | 83.02±12.63 | 86.03±9.55 |
| (2-Classes) | **TimesFM + MLP** | 83.75±3.15 | 83.71±3.31 | 83.02±2.68 | **83.16**±**3.04** | 89.63±1.92 | 87.95±4.06 |
| | **TimesFM + MLP + LoRA** | 81.14±4.26 | 82.04±2.59 | 80.94±3.66 | 80.48±4.22 | 90.20±1.87 | 90.00±2.11 |
| | **TimesFM + Attn** | 65.90±1.94 | 67.33±1.90 | 66.41±1.95 | 64.97±1.55 | 72.77±3.47 | 68.97±6.04 |
| | **FORMED** | **85.10**±**9.10** | **87.34**±**7.68** | 83.11±13.18 | 82.45±12.6 | **91.22**±**8.71** | **92.51**±**7.60** |
| | **MiniRocket** | 86.40±0.56 | 90.06±0.51 | 83.26±0.65 | 84.81±0.66 | — | — |
| | **PatchTST + FORMED** | 89.00±1.15 | 89.68±1.46 | 89.00±1.15 | 88.95±1.13 | 96.76±1.27 | 96.92±1.18 |
| TDBrain | **TimesFM + CNN** | 83.89±7.04 | 85.00±7.89 | 81.71±7.11 | 82.49±7.24 | 92.22±5.99 | 87.96±10.38 |
| (2-Classes) | **TimesFM + MLP** | **93.60**±**1.05** | **93.64**±**0.63** | **93.11**±**1.63** | **93.29**±**1.17** | 96.14±1.71 | 92.80±3.30 |
| | **TimesFM + MLP + LoRA** | 90.57±1.61 | 90.48±2.07 | 90.05±1.35 | 90.16±1.59 | 93.86±2.30 | 87.79±4.98 |
| | **TimesFM + Attn** | 86.87±1.26 | 90.05±0.60 | 83.98±1.84 | 85.42±1.62 | **96.43**±**1.62** | **95.34**±**1.77** |
| | **FORMED** | 87.47±8.38 | 87.58±8.14 | 86.95±10.20 | 86.38±9.34 | 94.20±5.52 | 87.62±10.82 |
| | **MiniRocket** | 93.46±0.51 | 91.61±1.17 | 84.14±1.35 | 87.27±1.08 | — | — |
| | **PatchTST + FORMED** | 90.29±0.65 | 88.19±3.23 | 71.61±4.96 | 76.03±3.97 | 94.79±0.64 | 98.98±0.27 |
| PTB | **TimesFM + CNN** | 94.05±1.23 | 90.17±2.29 | 86.84±3.14 | 88.36±2.65 | 94.22±3.39 | 98.29±1.06 |
| (2-Classes) | **TimesFM + MLP** | 87.12±0.43 | 77.20±0.82 | 76.70±1.12 | 76.92±0.67 | 76.80±1.11 | 91.45±0.39 |
| | **TimesFM + MLP + LoRA** | 92.76±2.04 | 87.80±4.03 | 85.80±3.30 | 86.66±3.21 | 85.83±3.27 | 94.70±1.39 |
| | **TimesFM + Attn** | 93.48±0.21 | 90.60±1.50 | 85.36±1.03 | 87.64±0.30 | 88.00±0.84 | 95.17±0.83 |
| | **FORMED** | **95.74**±**2.40** | **94.35**±**4.26** | **91.03**±**5.59** | **92.53**±**4.98** | **97.05**±**2.80** | **99.05**±**0.71** |
| | **MiniRocket** | 73.33±0.17 | 68.29±0.19 | 58.64±0.16 | 60.76±0.15 | — | — |
| | **PatchTST + FORMED** | 62.14±1.30 | 55.51±1.53 | 45.21±2.76 | 46.04±3.30 | 83.06±0.25 | 52.97±0.99 |
| PTB-XL | **TimesFM + CNN** | 71.08±0.49 | 62.73±0.90 | 58.28±0.44 | 59.97±0.41 | 88.49±0.37 | 63.48±0.99 |
| (5-Classes) | **TimesFM + MLP** | 65.22±1.12 | 55.26±1.32 | 52.63±0.71 | 53.46±0.82 | 81.35±0.48 | 51.24±1.01 |
| | **TimesFM + MLP + LoRA** | 67.10±0.64 | 58.04±1.15 | 54.81±0.58 | 55.29±0.79 | 85.04±0.49 | 57.07±0.52 |
| | **TimesFM + Attn** | 72.51±0.38 | 64.19±0.90 | 59.67±0.18 | 61.13±0.24 | 89.31±0.10 | 65.72±0.43 |
| | **FORMED** | **77.72**±**7.10** | **71.75**±**9.67** | **69.06**±**11.37** | **70.12**±**10.82** | **91.72**±**3.42** | **73.95**±**10.11** |

Table 7: **Full ablation study results of FORMED on all datasets.** Best results are **bolded**.

| Datasets | PE | CE | MoE | Accuracy | Precision | Recall | F1 score | AUROC | AUPRC |
|---|---|---|---|---|---|---|---|---|---|
| **ADFTD** (3-Classes) | ✓ | ✗ | ✓ | $48.81_{\pm2.04}$ | $42.92_{\pm1.75}$ | $43.67_{\pm1.04}$ | $42.54_{\pm1.09}$ | $61.43_{\pm1.06}$ | $43.83_{\pm1.24}$ |
| | ✗ | ✓ | ✓ | $53.77_{\pm3.04}$ | $47.80_{\pm1.50}$ | $47.55_{\pm0.73}$ | $46.26_{\pm0.76}$ | $65.46_{\pm0.50}$ | $48.20_{\pm0.29}$ |
| | ✗ | ✗ | ✓ | $45.87_{\pm2.24}$ | $40.83_{\pm1.54}$ | $42.65_{\pm1.75}$ | $40.57_{\pm1.85}$ | $60.43_{\pm2.48}$ | $42.17_{\pm2.44}$ |
| | ✓ | ✓ | ✗ | $54.08_{\pm2.15}$ | $48.61_{\pm2.26}$ | $48.16_{\pm1.33}$ | $46.78_{\pm1.96}$ | $66.43_{\pm2.00}$ | $49.52_{\pm2.96}$ |
| | ✓ | ✓ | ✓ | $\mathbf{66.83_{\pm18.35}}$ | $\mathbf{63.54_{\pm21.37}}$ | $\mathbf{65.25_{\pm21.63}}$ | $\mathbf{63.66_{\pm21.67}}$ | $\mathbf{77.70_{\pm16.16}}$ | $\mathbf{66.00_{\pm20.89}}$ |
| **APAVA** (2-Classes) | ✓ | ✗ | ✓ | $64.22_{\pm6.04}$ | $64.23_{\pm4.85}$ | $64.27_{\pm4.86}$ | $63.64_{\pm5.58}$ | $69.96_{\pm5.47}$ | $67.50_{\pm7.82}$ |
| | ✗ | ✓ | ✓ | $67.49_{\pm6.40}$ | $66.84_{\pm6.92}$ | $65.95_{\pm5.89}$ | $66.07_{\pm6.12}$ | $70.92_{\pm4.98}$ | $71.62_{\pm6.65}$ |
| | ✗ | ✗ | ✓ | $69.08_{\pm4.39}$ | $68.74_{\pm4.63}$ | $67.93_{\pm3.50}$ | $67.89_{\pm3.78}$ | $73.25_{\pm2.44}$ | $73.52_{\pm2.43}$ |
| | ✓ | ✓ | ✗ | $65.54_{\pm4.25}$ | $65.39_{\pm4.38}$ | $64.84_{\pm4.03}$ | $64.50_{\pm4.20}$ | $69.52_{\pm5.29}$ | $70.18_{\pm6.97}$ |
| | ✓ | ✓ | ✓ | $\mathbf{85.10_{\pm9.10}}$ | $\mathbf{87.34_{\pm7.68}}$ | $\mathbf{83.11_{\pm13.18}}$ | $\mathbf{82.45_{\pm12.6}}$ | $\mathbf{91.22_{\pm8.71}}$ | $\mathbf{92.51_{\pm7.60}}$ |
| **TDBrain** (2-Classes) | ✓ | ✗ | ✓ | $\mathbf{88.08_{\pm3.24}}$ | $\mathbf{88.83_{\pm2.95}}$ | $86.57_{\pm4.14}$ | $\mathbf{87.23_{\pm3.61}}$ | $96.11_{\pm2.24}$ | $92.80_{\pm4.31}$ |
| | ✗ | ✓ | ✓ | $87.66_{\pm5.12}$ | $88.68_{\pm4.68}$ | $85.87_{\pm6.26}$ | $86.63_{\pm5.75}$ | $96.05_{\pm2.58}$ | $92.39_{\pm5.67}$ |
| | ✗ | ✗ | ✓ | $86.13_{\pm2.47}$ | $87.73_{\pm2.00}$ | $83.81_{\pm3.15}$ | $84.87_{\pm2.97}$ | $\mathbf{96.58_{\pm0.76}}$ | $\mathbf{94.49_{\pm1.05}}$ |
| | ✓ | ✓ | ✗ | $87.57_{\pm4.82}$ | $87.81_{\pm5.26}$ | $86.42_{\pm5.04}$ | $86.87_{\pm5.03}$ | $95.85_{\pm3.93}$ | $93.04_{\pm6.98}$ |
| | ✓ | ✓ | ✓ | $87.47_{\pm8.38}$ | $87.58_{\pm8.14}$ | $\mathbf{86.95_{\pm10.20}}$ | $86.38_{\pm9.34}$ | $94.20_{\pm5.52}$ | $87.62_{\pm10.82}$ |
| **PTB** (2-Classes) | ✓ | ✗ | ✓ | $93.70_{\pm0.63}$ | $90.01_{\pm2.22}$ | $87.17_{\pm1.27}$ | $88.42_{\pm0.88}$ | $95.65_{\pm0.51}$ | $98.88_{\pm0.30}$ |
| | ✗ | ✓ | ✓ | $89.86_{\pm1.05}$ | $82.24_{\pm2.65}$ | $82.05_{\pm0.74}$ | $82.03_{\pm1.04}$ | $91.48_{\pm0.65}$ | $97.71_{\pm0.44}$ |
| | ✗ | ✗ | ✓ | $93.23_{\pm0.78}$ | $88.65_{\pm1.78}$ | $87.02_{\pm2.66}$ | $87.70_{\pm1.61}$ | $95.76_{\pm0.82}$ | $98.93_{\pm0.29}$ |
| | ✓ | ✓ | ✗ | $90.92_{\pm0.76}$ | $84.41_{\pm1.46}$ | $81.94_{\pm2.04}$ | $83.00_{\pm0.90}$ | $91.47_{\pm3.26}$ | $97.63_{\pm1.20}$ |
| | ✓ | ✓ | ✓ | $\mathbf{95.74_{\pm2.40}}$ | $\mathbf{94.35_{\pm4.26}}$ | $\mathbf{91.03_{\pm5.59}}$ | $\mathbf{92.53_{\pm4.98}}$ | $\mathbf{97.05_{\pm2.80}}$ | $\mathbf{99.05_{\pm0.71}}$ |
| **PTB-XL** (5-Classes) | ✓ | ✗ | ✓ | $70.03_{\pm1.86}$ | $61.82_{\pm1.90}$ | $55.51_{\pm3.32}$ | $56.55_{\pm3.92}$ | $87.22_{\pm1.72}$ | $61.63_{\pm3.12}$ |
| | ✗ | ✓ | ✓ | $66.32_{\pm2.99}$ | $57.68_{\pm4.91}$ | $51.96_{\pm5.84}$ | $52.60_{\pm5.90}$ | $85.38_{\pm2.94}$ | $57.92_{\pm4.99}$ |
| | ✗ | ✗ | ✓ | $71.53_{\pm0.71}$ | $63.31_{\pm0.84}$ | $57.63_{\pm1.73}$ | $59.19_{\pm1.51}$ | $88.61_{\pm0.66}$ | $64.38_{\pm1.33}$ |
| | ✓ | ✓ | ✗ | $68.66_{\pm1.24}$ | $59.89_{\pm1.29}$ | $55.32_{\pm1.45}$ | $56.16_{\pm1.63}$ | $87.28_{\pm0.74}$ | $61.00_{\pm1.48}$ |
| | ✓ | ✓ | ✓ | $\mathbf{77.72_{\pm7.10}}$ | $\mathbf{71.75_{\pm9.67}}$ | $\mathbf{69.06_{\pm11.37}}$ | $\mathbf{70.12_{\pm10.82}}$ | $\mathbf{91.72_{\pm3.42}}$ | $\mathbf{73.95_{\pm10.11}}$ |

