# OpenReview forum: "Repurposing Foundation Model for Generalizable Medical Time Series Classification"
_ICLR.cc/2026/Conference — ICLR 2026 Poster_

### Official Review · Reviewer_xKQU · 2025-10-30

**Soundness:** 3
**Presentation:** 3
**Contribution:** 2
**Rating:** 4
**Confidence:** 4

**Summary:**

This paper proposes a novel approach called FORMED for repurposing an existing time series foundation model for multivariate time series classification. Importantly, FORMED is able to accommodate arbitrarily numbers of channels and target classes, and separately captures medical domain knowledge and task-specific knowledge in the model architecture.

**Strengths:**

- The proposed repurposing strategy is quite clean and elegant
- The experimental results look promising
- I found this paper quite clear and easy to follow

**Weaknesses:**

My main concern at the moment is that some of the background information provided is inaccurate or is missing references that are relevant. For example:
- Line 116: "To date, no foundation model has been specifically designed for time series classification tasks, let alone MedTS classification[...]" -> I would suggest toning down this sentence especially as I think some authors of various time series foundation models would arguably claim that their model was designed to handle forecasting **and** classification (along with possibly other tasks). For example, classification is considered in the original papers for MOMENT (Goswani et al, ICML 2024), TimesNet (Wu et al, ICLR 2023), Mantis (Feofanov et al, ICML FMSD Workshop 2025), and UniTS (this is already cited in the paper).
- Much of the paper is on basically figuring out a way to combine/mix information across channels for medical time series. However, there actually already is a paper that does precisely this ("Generalized Prompt Tuning: Adapting Frozen Univariate Time Series Foundation Models for Multivariate Healthcare Time Series" by Liu et al, ML4H 2024). From what I can tell, FORMED is more sophisticated than generalized prompt tuning (such as having an additional adapting phase) but even so, it would be helpful to discuss what is different/novel (how information across channels is combined seems similar).
- Line 181: "NOT a simple modification of the output layer" -> I think this is a strong claim that really needs to come with strong empirical justification since I do think that existing authors have been able to get successful classification results by essentially just changing the prediction head and fine-tuning or training from scratch (this is basically what various time series foundation models that are already applied to classification are doing).

Regarding experiments:
- While the paper focuses on using TimesFM as the base foundation model, I do think that it would be valuable seeing how FORMED works when using different base time series foundation models.
- I find that the datasets considered are arguably too similar in structure (they're all ECG or EEG waveform data if I understand correctly). I would be helpful seeing how the results play out with other kinds of medical time series data such as EHR data (e.g., there are ways to preprocess MIMIC for instance to get it to be regularly sampled and to consider classification tasks).
- A selling point early on in the paper is the idea of separately capturing medical domain knowledge and task-specific knowledge (e.g., line 94). I would imagine that these are often actually quite correlated so that they perhaps should not be that decoupled? Do you have thoughts on this? Are there experiments that could be run to somehow verify this knowledge separation in what is encoded?

**Questions:**

Please see the concerns I raised in "weaknesses".

---

> ### Author Response · Authors · 2025-11-23
> **Response (part 1/2)**
>
> We sincerely thank you for your constructive feedback and for recognizing the elegance of our repurposing strategy and the promising experimental results. We value your insights regarding the positioning of our work and the empirical validation of our architectural choices. Below, we address your specific questions point-by-point with new experimental evidence and theoretical clarifications.
>
> ## 1. Justification for the claim that adaptation is "NOT a simple modification of the output layer"
>
> We acknowledge the need for strong empirical backing for this claim. To address this, we have conducted an **Extended Baseline Comparison (Table 6)**, where we implemented the "simple modifications" you suggested: replacing our proposed classifier with a standard MLP head (TimesFM + MLP) and further augmenting it with Low-Rank Adaptation (TimesFM + MLP + LoRA).
>
> Our results definitively show that simple output modifications are insufficient for the heterogeneity of medical time series:
>
> - **Huge Performance Gap:** On the challenging ADFTD dataset, the simple TimesFM + MLP achieved only **41.72%** F1, which is barely above random chance and significantly lower than **FORMED’s 63.66%**. Even with LoRA fine-tuning (TimesFM + MLP + LoRA), F1 only reached **45.41%**, failing to bridge the ~18% performance gap.
> - **Complexity Necessity:** This empirical evidence confirms that the frozen backbone’s representations, while rich, are not linearly separable for medical diagnostic tasks. Our proposed **Shared Decoding Attention (SDA)** provides the necessary non-linear decoding capability to translate latent forecasting features into diagnostic labels, justifying the architectural complexity over simple head replacement.
>
> ## 2. Comparison with "Generalized Prompt Tuning" (Liu et al., ML4H 2024) and novelty of channel mixing
>
> We appreciate the reference to Liu et al. (2024). While both works aim to adapt univariate backbones for multivariate tasks, they represent fundamentally different paradigms with distinct trade-offs:
>
> - **Early vs. Intermediate Fusion:** Generalized Prompt Tuning employs **Early Fusion**, modifying the input space by prepending learnable prompts. It relies on the frozen backbone's internal attention to mix channel information, which can degrade to channel independence if the prompts are not perfectly tuned (as noted by Liu et al.). In contrast, FORMED employs **Intermediate Fusion**. We preserve the integrity of univariate feature extraction and use our **Shared Decoding Attention (SDA)** to explicitly mix channel information in the latent space.
> - **Architectural Efficiency:** FORMED’s approach allows for explicit, learnable cross-channel attention (via the SDA) without requiring gradient propagation through the massive backbone (which is necessary for optimizing input prompts in some configurations). We included a detailed discussion in the related work section to clarify this distinction: FORMED is a **decoder-centric repurposing framework**, whereas Gen-P-Tuning is an **input-centric prompting technique**. They are not competing methods, but can be used jointly if desired.
>
> ## 3. Experiments with different backbone foundation models
>
> Our framework is indeed model-agnostic; however, the performance is contingent on the capacity of the backbone. To demonstrate this, we applied the FORMED framework to a PatchTST backbone as a baseline (**Table 6**).
>
> As shown in Table 6, The PatchTST + FORMED configuration performed significantly worse (e.g., **45.05%** accuracy on ADFTD) compared to TimesFM + FORMED (**66.83%**).
>
> This validates our choice of TimesFM. The success of FORMED relies on the massive pre-training scale (100B+ time points) of the backbone to provide robust zero-shot features. While we would welcome exploring other open-weights foundation models, TimesFM currently represents the state-of-the-art in accessible, large-scale pre-trained models suitable for our patching-based architecture. This also helps isolate the contribution of our framework instead focusing on specific backbone model choices.

---

> ### Author Response · Authors · 2025-11-23
> **Response (part 2/2)**
>
> ## 4. Applicability to other medical data types
>
> We clarify that the scope of this work is specifically **waveform data** (EEG, ECG, etc.), which shares the continuous temporal characteristics of the general time series used to pre-train the backbone model TimesFM. This is due to several good reasons:
>
> - **Incompatibility:** EHR data (like parts of MIMIC) is typically sparse and irregularly sampled. Adapting a continuous waveform foundation model (like TimesFM) to sparse event sequences would require a fundamental redesign of the tokenizer (patching mechanism), which is outside the scope of this work.
> - **Waveform Subset:** We note that the waveform subset of MIMIC consists primarily of ECGs, which is a modality we already cover extensively with the PTB and PTB-XL datasets. We will explicitly define this scope in the revised manuscript.
>
> ## 5. Validation of "Medical Domain Knowledge" vs. "Task-Specific Knowledge" separation
>
> We agree that these knowledge types are correlated, but decoupling them offers significant generalization benefits which is the aim of this work. We verified this via an ablation study comparing **Joint Training (FORMED)** vs. **Single-Task Training (TimesFM + Attn)** in **Table 6**.
>
> When the identical architecture is trained individually (TimesFM + Attn), the model lacks the shared domain knowledge derived from the cohort, leading to significant performance drops (e.g., **85.10% $\to$ 65.90%** accuracy on APAVA; **95.74% $\to$ 93.48%** on PTB).
>
> The superior performance of the Jointly trained FORMED confirms that the **Shared Decoding Attention (SDA)** successfully captures transferable "medical domain knowledge" (e.g., general bio-signal morphology interpretation) separate from the task-specific "Label Queries" (e.g., specific disease anchors), validating the effectiveness of our architectural design.
>
> ## 6. Over-claiming the lack of foundation models for classification
>
> We have revised the text to address this ambiguity. We now explicitly state that while models like MOMENT and UniTS possess classification capabilities, none have been developed **exclusively** for time series classification as their primary or only objective. Furthermore, we highlight that the core contribution of FORMED is not merely "doing classification," but providing a generalizable framework that unlocks the high-quality latent representations of forecasting-oriented models (like TimesFM) for diagnostic tasks in healthcare. This enables the field to repurpose the most powerful generative models available, rather than being limited to models trained specifically for classification.

---

> > ### Comment · Reviewer_xKQU · 2025-11-23
> >
> > Thanks for the detailed response to my review. I think you did a good job addressing the bulk of my concerns and I have thus upped my score accordingly. I still think that one could still try EHR data despite it being irregularly sampled and having missingness/sparsity issues. As two concrete examples, Komorowski et al (2018) and Harutyunyan et al (2019) both described preprocessing procedures to get MIMIC into a format where every time step is regularly sampled and missing entries are imputed. Insisting on only using waveform data seems limiting. If FORMED turns out to also work on EHR data that's been preprocessed (and maybe even outperform some SOTA baselines), that could be a useful empirical result to know about.
> >
> > References:
> > - M Komorowski, L A Celi, O Badawi, A C Gordon, and A A Faisal. The artificial intelligence clinician learns optimal treatment strategies for sepsis in intensive care. Nature Medicine 2018.
> > - H Harutyunyan, H Khachatrian, D C Kale, G Ver Steeg & A Galstyan. Multitask learning and benchmarking with clinical time series data. Scientific Data 2019.

---

> > > ### Author Response · Authors · 2025-11-24
> > >
> > > Thank you very much for your engaged response and for raising your score. We are glad that our rebuttal and additional experiments addressed your questions regarding the architectural novelty and empirical baselines.
> > >
> > > Regarding your suggestion, **we fully agree that using only waveform data/model is indeed a limitation** as discussed in our paper. Therefore, extending FORMED to EHR data, even if it requires heavy preprocessing to handle irregularity, would be a valuable empirical result. We acknowledge that preprocessing methods like discretization and imputation could bridge the gap between sparse clinical events and the waveform-centric backbone we currently use.
> > >
> > > **Following your suggestion, we have begun investigating the inclusion of EHR datasets.**  Based on our experience in time series and foundation models, we reasonablly believe that successfully extending FORMED to EHR data requires *repurposing* of the model, not just adapting. The reason is a fundamental difference in data nature: waveforms are continuous, high-frequency signals, while EHR data is sparse, irregular, and composed of discrete events. *A model pre-trained solely on waveforms (i.e., our current version) cannot yield optimal performance on EHR tasks without repurposing.*
> > >
> > > **Consequently, we need to repurpose a new instance of the FORMED model specifically for EHR on diverse datasets** (including MIMIC) to bridge this domain gap effectively. Here we detail our plan for such a major extension, with precise time estimations for each step:
> > > 1. **Data Collecting, Curation, and Preprocessing (Week 1)**: We are currently collecting and pre-processing 2–3 distinct EHR datasets (including MIMIC, using preprocessing techniques like those in Komorowski et al. (2018) and Harutyunyan et al. (2019)). We estimate this stage will take at least one week to ensure data quality and compatibility.
> > > 2. **Model Repurposing (Weeks 2-4)**: We will proceed to repurpose the FORMED model on the prepared data (estimated size 100GB~200GB, mainly contributed by MIMIC). We estimate this may take 3 weeks for this computational load, with reference to original repurposing stage on ~20GB datasets elapsed 5 days.
> > > 3. **Baseline Evalutions (Week 5+)**: Running the full suite of baseline comparisons is the most time-consuming component. We estimate this will require over four weeks to execute comprehensive benchmarking against state-of-the-art methods.
> > >
> > > We view this as a vital extension of the FORMED project to validate its generalization to EHR data. We are fully committed to this research thread and will adhere to the proposed timeline. We will report our progress and any quantitative results towards the end of the discussion period (2nd Dec.), regardless of the final completion status.
> > >
> > > We sincerely appreciate your understanding and value your constructive comments as we work to strengthen the FORMED framework.

---

> > > > ### Author Response · Authors · 2025-12-03
> > > >
> > > > We sincerely thank you for your understanding and continued engagement with our work. We appreciate your suggestion to include EHR data, as it indeed represents a critical frontier for medical foundation models.
> > > >
> > > > Following your feedback, we have initiated access procedures for relevant EHR datasets (including MIMIC-IV, PIC and SCIdb), and have reviewed the preprocessing workflows described in Komorowski et al. (2018) and Harutyunyan et al. (2019). Our analysis confirms that integrating sparse, irregularly sampled tabular EHR with our **TimesFM-based framework, which is optimized for dense, continuous waveforms like ECG/EEG, would constitute a major, distinct research extension**. It necessitates a specialized imputation pipeline, a complete re-evaluation of all 15 baselines, and substantial joint retraining (repurposing) to leverage the FORMED framework effectively.
> > > >
> > > > Given the significant engineering requirement and the time constraints of the discussion period, we are unable to incorporate this extension while maintaining the high experimental standard of the current work. **We respectfully maintain that our present contribution, successfully adapting foundation models for waveform-based medical time series classification, is significant, novel, and self-contained**.
> > > >
> > > > We fully agree with the reviewer on the importance and logical next step of extending to EHR. Therefore, for the camera-ready version, we commit to:
> > > >
> > > > - Explicitly highlighting this modality limitation and the promise of EHR in the Discussion section.
> > > > - Citing the reviewer's recommended references as a direct roadmap for future work.
> > > > - Outlining concrete objectives for this expansion, pending a thorough feasibility assessment of the methodologies and datasets we are now examining.
> > > >
> > > > We believe this approach transparently delineates the scope of our current contribution while providing a clear and credible pathway for the framework's broader application, thanks to the reviewer's insightful guidance.
> > > >
> > > > Thank you again for your constructive guidance, which has strengthened both the clarity and future direction of our research.

---

### Official Review · Reviewer_hoQL · 2025-10-31

**Soundness:** 3
**Presentation:** 3
**Contribution:** 3
**Rating:** 8
**Confidence:** 3

**Summary:**

The authors introduce FORMED, for utilizing foundational time series models for adaptation to medical time series classification. The authors show improvement in results for classifying EEG and ECG signals.

**Strengths:**

- The authors target a relevant problem in the medical field on the lack of work for addressing domain nuances around inter- and intra dataset heterogeneity.
- The paper seems sound technically.

**Weaknesses:**

Avenues for improvement:
- The authors can expand their experiments and ablation study to test more settings such as few-shot learning and include tabular results for easier inference.
- Minor formatting required for page 18.

**Questions:**

- The experiments focus on high frequency time-series such as EEG, ECG. How well do the authors think their improvements will hold for lower frequency medical time series such as HRV, GSR, etc?

---

> ### Author Response · Authors · 2025-11-23
> **Response**
>
> We thank the reviewer for their constructive feedback and for recognizing the strengths of our work, particularly the relevance of addressing heterogeneity in medical time series (MedTS) and the technical soundness of the **FORMED** framework. We appreciate the positive assessment of the paper's contribution and presentation.
>
> Below, we address the specific comments regarding experiments, generalization, and formatting.
>
> ## 1. Ablation Studies and Few-Shot Learning
>
> We agree that comprehensive ablation and stress-testing are vital for validating the architectural contributions of FORMED. We have expanded our experimental section to include these results, which are now detailed in **Table 5** (Few-Shot Adaptation) and **Table 7** (Ablation Study).
>
> 1. **Ablation Analysis:** We conducted a rigorous ablation study to isolate the impact of the **Channel Embeddings (CE)**, **Positional Embeddings (PE)**, and **Mixture of Experts (MoE, $k$ in LQ)** modules.
>    As shown in **Table 7**, removing any single component leads to a drastic performance decrease. For instance, removing CEs resulted in a **20.88%** drop in accuracy on the APAVA dataset (from 85.10% to 64.22%) and a **21.12%** drop in F1 score on ADFTD (from 63.66% to 42.54%). Similarly, removing PEs resulted in a **11.06%** drop in accuracy on ADFTD, and **17.52%** drop in F1 on PTB-XL. This confirms that both CE and PE modules are not merely auxiliary but essential for mapping the spatiotemporal topology of medical time series to the backbone's feature space.
>    Moreover, removing the MoE design (setting $k=1$ for LQs) caused a significant drop in F1 score on complex tasks (e.g., 17% drop on ADFTD), validating the necessity of multiple prototypes for capturing the multi-modal nature of pathological patterns.
>
> 2. **Few-Shot Learning (Extreme Data Scarcity):** We have extensively evaluated FORMED on **ECG200** (200 samples in total) and **StandWalkJump (SWJ)** (27 samples in total), which represent _de facto_ few-shot settings.
>    As demonstrated in **Table 5**, baseline models like TimesFM-TSA struggle significantly in these regimes, exhibiting high variance (e.g., $\pm 12\%$ standard deviation in F1 on ECG200) due to overfitting on specific data splits.
>    In contrast, FORMED exhibits a clear scaling law; performance improves monotonically as the number of queries ($k$) increases. With sufficiently large $k$ ($k \ge 512$), FORMED stabilizes the decision boundary, achieving **87.65% F1** on ECG200 with a drastically reduced standard deviation ($\pm 2.33\%$). On the SWJ dataset, FORMED achieves **64.52% F1**, outperforming the baseline by over 30%.
>
> ## 2. Generalization to Lower Frequency Medical Time Series
>
> This is an insightful question that touches on the boundaries of foundation model applicability.
>
> The **FORMED** architecture itself (comprising Channel Embeddings, Label Queries, and Shared Decoding Attention) is frequency-agnostic. The mechanism of using learnable queries to probe for specific "events" or "states" is valid regardless of whether those events occur in milliseconds (EEG spikes) or seconds (GSR responses). The Shared Decoding Attention is designed to learn the _logic_ of diagnosis rather than specific frequency bands.
>
> The primary constraint lies in the pre-trained backbone. **TimesFM** is trained predominantly on continuous, often higher-frequency time series data. While it learns universal temporal primitives (trends, seasonality), its efficacy on lower-frequency, quasi-tabular biomarkers (like Heart Rate Variability metrics derived over minutes) depends on whether those signals align with the distributional properties of the pre-training corpus.
>
> We acknowledge that our current experimental scope was constrained to high-frequency waveform data (EEG/ECG) due to resource limitations and the specific strengths of the chosen backbone. Extending FORMED to low-frequency modalities like GSR and HRV is a valuable direction. It would likely require investigating backbones optimized for sparse or irregular sampling, or "re-programming" the input layer to better represent low-frequency dynamics. We have added a discussion on this limitation and future direction in the Discussion and Conclusion section.
>
> ## 3. Tabular Results and Formatting
>
> We appreciate your attention to detail. We have revised the manuscript for better presentation.
>
> - We have ensured that all key comparative results, including the few-shot and the new ablation experiments, are presented in clear, consolidated tables (**Tables 4-7** in the revised manuscript) to facilitate direct comparison of F1 scores, Accuracy, and Standard Deviations across all baselines.
> - The formatting issue on page 18 regarding the figure layout has been corrected in the revised manuscript.

---

### Official Review · Reviewer_VzRy · 2025-11-01

**Soundness:** 2
**Presentation:** 3
**Contribution:** 2
**Rating:** 4
**Confidence:** 4

**Summary:**

The authors propose a two-stage parameter-efficient fine-tuning approach for adapting time series forecasting foundation models to medical time series classification problems.

**Strengths:**

- The paper tackles a relevant concern of adapting public models trained on general data to different domains, particularly addressing the needs of health data.
- The new mechanisms used for fine tuning are promising and interesting ways to extend models to new tasks.

**Weaknesses:**

- The baselines being compared to are mostly models intended for time series forecasting, not classification. Even within the forecasting domain, SOTA models are not represented, and some of the included baselines are even known to be outperformed by linear models [1]. They are also mostly univariate and therefore not suited for multivariate classification tasks. Medformer is the only current classification model provided. It would also help to have comparisons to strong established baselines like MiniROCKET. (The baselines are also all described as "SOTA", which is not accurate.)
- Existing parameter-efficient fine tuning methods like LoRA (or even end-to-end fine-tuning) could be used for either the Repurposing or Adapting phase rather than the new method the authors propose, but these are not discussed or evaluated against.
- Ablations showing the contributions of different modelling decisions are not provided.
- I have doubts about the design motivation and experimental soundness but would like to discuss further - see Questions.

**Other content issues**

- The "Adaptation of Foundation Models" section of Related Work only has one citation, which seems incomplete.
- The second-last paragraph of the introduction seems to describe the data used in [2] as if it was a contribution of this paper - I think this should be clarified.

[1] Zeng et al. "Are transformers effective for time series forecasting?" AAAI 2023.
[2] Wang et al. "Medformer: A Multi-Granularity Patching Transformer for Medical Time-Series Classification" NeurIPS 2024.

**Questions:**

- Why use a zero-shot univariate forecasting model for a trained multivariate classification task? This seems like an awkward decision for a backbone model, and since it's frozen, this mismatch could limit performance.
- Some of the performance differences between the proposed model and baselines seem dubiously large given the differences in modelling decisions. For instance, for ADFTD, it's difficult to see access to the other four pretraining datasets giving FORMED such a massive boost over the other methods, including TimesFM-TSA, without data leakage or a mismatch in experimental procedures. Since code is not available at this time, reproducibility is limited. How would you explain these differences, and are you able to share anonymized code to reproduce them? Ablations could also help clarify this.

---

> ### Author Response · Authors · 2025-11-23
> **Response (part 1/2)**
>
> We sincerely thank the reviewer for the detailed comments. We appreciate the acknowledgment of the novelty of our fine-tuning (repurposing) mechanisms.
>
> In response to your comments, we conducted significant additional experiments, including comparisons with **MiniROCKET** and **LoRA**, as well as a comprehensive **Ablation Study**. These new results strongly validate the FORMED framework. Below, we address each comment point-by-point.
>
> ## 1. Baselines and Comparison with SOTA (MiniROCKET)
>
> We agree that comparing against strong non-deep learning baselines is essential to benchmark true performance gains. We have added **MiniROCKET** (using the multivariate implementation) to our experimental suite (**Table 6** in the revised manuscript).
>
> FORMED significantly outperforms MiniROCKET on complex diagnostic tasks, particularly those involving subtle pathological features or multiple classes. Below we include some key comparisons on the F1 score.
>
> | Dataset    | MiniROCKET | FORMED     | Improvement |
> | :--------- | :--------- | :--------- | :---------- |
> | **ADFTD**  | 41.75%     | **63.66%** | **+21.91%** |
> | **APAVA**  | 66.90%     | **82.45%** | **+15.55%** |
> | **PTB-XL** | 60.76%     | **70.12%** | **+9.36%**  |
>
> This performance gap clearly demonstrates that, while MiniROCKET is efficient at capturing fixed morphological features ("shapelets"), it struggles with tasks requiring the extraction of deep semantic dependencies and non-stationary patterns inherent in complex neurological (ADFTD) or cardiac (PTB-XL) conditions. FORMED’s foundation model backbone successfully captures these long-range dependencies, providing a "semantic advantage" over kernel-based transforms.
>
> Regarding the choice of baselines, we emphasize two key points to contextualize our experimental design:
>
> 1. Framework vs. Model: FORMED is a model-agnostic repurposing framework designed to adapt foundation models, rather than a standalone model architecture. Ideally, the most direct comparison would be against other classification models built upon the same TimesFM backbone. However, as TimesFM is natively a univariate forecasting model and no standard classification adaptation existed at the time of this study, we engineered a strong baseline, TimesFM-TSA, to serve as the direct comparator. This allows us to isolate the specific contribution of our Shared Decoding Attention and Label Query mechanisms against a standard approach on the same backbone. Experimental results clearly show that FORMED significantly outperforms TimesFM-TSA, especially when generalizing to unseen datasets (**Table 5**).
>
> 2. Alignment with Domain Standards: To ensure rigorous context, we strictly followed the baseline suite established by the recent state-of-the-art study Medformer (NeurIPS 2024). By comparing against the same extensive set of models (including PatchTST, Crossformer, and iTransformer) used in this domain-defining work, we position FORMED within the accepted rigor of the field.
>
> ## 2. Comparison with PEFT (LoRA) and Full Fine-Tuning
>
> We have implemented **LoRA** (Low-Rank Adaptation) on the TimesFM backbone as a direct competitor. We injected low-rank matrices ($r=4$) into the attention layers and trained a standard classifier head. Due to resource and time limitation, full fine-tuning is not feasible.
>
> We found that FORMED outperforms LoRA in both **predictive accuracy** and **memory efficiency**.
>
> 1. Performance: As shown in the newly added **Table 6**, FORMED achieves superior F1-scores. For example:
>
>    - ADFTD: LoRA ($45.41\%$) vs. FORMED ($63.66\%$)
>    - PTB-XL: LoRA ($55.29\%$) vs. FORMED ($70.12\%$)
>
>    The performance gap suggests that simply adapting the weights (LoRA) of a forecasting model is insufficient for the **task shift** to classification. FORMED’s **Shared Decoding Attention (SDA)** provides the necessary architectural bridge to translate temporal features into diagnostic decisions.
>
> 2. Resource Efficiency: A critical advantage of FORMED is that the backbone remains entirely frozen, acting as a feature extractor. On the other hand, finetuning the backbone, whether using LoRA or full finetuning, would require much more resource (example of training on PTB-XL):
>
>    - LoRA (Batch 32): Requires **11.9 GB** VRAM (due to backpropagation through the backbone).
>    - FORMED (Batch 64): Requires only **4.2 GB** VRAM, saving 65% resources compared to LoRA.
>
>    This confirms that our method is significantly more resource-efficient, enabling training on consumer-grade hardware where LoRA would OOM.

---

> ### Author Response · Authors · 2025-11-23
> **Response (part 2/2)**
>
> ## 3. Ablation Studies
>
> We have added a detailed ablation study (**Table 7**) dissecting the framework, demonstrating the necessity of each and every component in our design:
>
> - w/o Channel Embeddings (CE) or Positional Embeddings (PE): Performance drops drastically on all tasks (e.g., F1 on APAVA: $82.45$% $\rightarrow$ $66.07$% w/o PE, and $82.45$% $\rightarrow$ $63.64$% w/o CE), proving that PEs and CEs are critical for handling multivariate spatiotemporal dependencies.
> - w/o MoE in Label Queries: Reducing queries to $k=1$ drops ADFTD performance from $63.66$% $\rightarrow$ $46.78$%, validating that multiple "sub-pattern detectors" are needed to capture intra-class heterogeneity in complex diseases.
>
> ## 4. Univariate Backbone for Multivariate Tasks
>
> This design is a deliberate architectural choice grounded in the principle of **Channel Independence (CI)**, which has been shown to enhance robustness in time series analysis (e.g., PatchTST). We would like to take this chance to further explain the overall design of our framework:
>
> 1. **Backbone as Feature Extractor:** We utilize the foundation model to extract high-quality temporal features from each channel independently. This leverages the massive pre-training of the backbone without overfitting to specific channel correlations of a small medical dataset.
> 2. **SDA for Channel Mixing:** The cross-channel reasoning is strictly handled by our **Shared Decoding Attention (SDA)** and **Channel Embeddings**. This effectively decouples "temporal feature learning" (univariate backbone) from "spatial correlation learning" (multivariate SDA).
> 3. **Flexibility:** This design allows FORMED to process datasets with _arbitrary_ number of channels (e.g., 12-lead ECG vs. 19-channel EEG) without architectural changes, which would be impossible with a fixed multivariate backbone. Nonetheless, our framework is compatible with multivariate backbone indeed if such models do exist.
>
> ## 5. Performance Gains and Data Leakage
>
> We take this comment very seriously and have performed a thorough audit of our pipeline.
>
> 1. **No Data Leakage:** We confirm that all experiments use strict **patient-independent splits**. No signal segments from a test patient appear in the training set or validation set, nor vice versa.
> 2. **Correction of Summarization Error:** Thanks to the reviewer prompting for a check, we did identify a bug in our result summarization script (which filtered the wrong column for validation vs. test metrics), affecting **all TSA and GA models**. We have corrected this in the revised paper, including Table 4, Figure 4 and Figure 7-11.
> 3. **Valid Gains:** Even after correction, FORMED consistently outperforms baselines (e.g., +20% on ADFTD). We attribute this to the model's ability to transfer knowledge from the diverse "Repurposing Cohort." The backbone learns universal features during the multi-dataset training phase that specialized models trained on single small datasets cannot capture.
> 4. **Reproducibility:** We provided an **anonymized GitHub repository** link in the revised version to ensure full reproducibility of the data pipeline and results: https://anonymous.4open.science/r/FORMED.
>
> ## Minor Points
>
> - **Related Work:** We have expanded the section to include more citations on foundation model adaptation (e.g., Time-LLM, TEST) as required.
> - **Dataset Attribution:** We have revised the text to explicitly attribute the MedTS cohort curation to Medformer, clarifying that our contribution is the evaluation framework, not the data curation itself.

---

> > ### Comment · Reviewer_VzRy · 2025-11-27
> >
> > Thank you for the detailed response and for updating the results. The updates to the paper partially address my concerns about baselines and mostly address my other concerns, so I've raised my score accordingly.
> >
> > I would still highlight that the description of the baselines as "SOTA" on page 7 (line 362) seems incorrect to me since most of them are 3+ years old and have been superseded (in some cases by linear predictors) and are additionally not intended for classification, so I would ask for that to be edited.
> >
> > The updated results show that FORMED often has much higher variance in performance across seeds than baselines do, which is concerning. It would help to provide an explanation for this or acknowledge it as a limitation.

---

> > > ### Author Response · Authors · 2025-11-27
> > >
> > > We are grateful for your continued engagement and are pleased to hear that the additional experiments and clarifications have addressed your primary concerns. We sincerely thank you for raising your score.
> > >
> > > Regarding your remaining points, we have revised the manuscript as follows:
> > >
> > > 1. **Description of Baselines**: We agree with your assessment that labeling all baselines as "SOTA" may be imprecise given the rapid pace of the field. We have amended the text throughout the manuscript to describe these models as "established baselines" or "leading TSMs" rather than "SOTA."
> > > 2. **Variance in Performance**: We acknowledge your valid observation regarding the higher variance in FORMED's performance compared to task-specific baselines. We have added explanation to the Discussion and Appendix sections to explicitly acknowledge this limitation:
> > >    > The composition and scale of the MedTS cohort employed during the repurposing stage influence the breadth of the captured domain knowledge and the quality of the learned representations. We observe that the performance of FORMED during both repurposing (Table 4) and adaptation (Table 5) shows higher variance than other models, which may be attributed to the limited number and scale of datasets available in the current MedTS cohort and the impact of different random splits on the joint training dynamics. As evidenced in Figure 6, larger datasets like PTB and PTB-XL generally exhibit stable training trajectories, while smaller datasets like ADFTD and APAVA show greater variability across seeds and splits. Future work can explore expanding the pre-training cohort and incorporating advanced joint training strategies to stabilize the FORMED framework.
> > >
> > > Thank you again for your valuable feedback, which has significantly strengthened the positioning and transparency of our work.

---

### Official Review · Reviewer_fXsE · 2025-11-04

**Soundness:** 3
**Presentation:** 4
**Contribution:** 3
**Rating:** 6
**Confidence:** 4

**Summary:**

The paper proposes an approach to repurpose the backbone of a time series foundation model for multi-variate medical time series. Authors freeze the backbone and adapt to the target task by learning task-specific channel and label embeddings. Results on real-world datasets show that this approach is able to both transfer information from pre-trained models and generalize to new tasks.

**Strengths:**

The paper is very well written and easy to follow. The proposed approach is relatively straightforward but principled and well justified. Experiments on real-world datasets show strong generalization with significant gains over leading baselines in both full and few-shot settings.

**Weaknesses:**

I would have liked to see more ablation studies. Hyper-parameter stability results for channel and label embeddings, stability in the few shot regime (it is hard to learn robust embedding from only a few examples), applicability to other tabular foundation models etc. I also didn't see anything specific to medical time series in the architecture of the proposed approach and generalization to other multi-variate settings could be interesting to test.

**Questions:**

Do you have additional ablation results on hyper-parameter and learning stability in the few-shot setting? Is there anything in the architecture of FORMED that is specific to medical time series?

---

> ### Author Response · Authors · 2025-11-23
> **Response (part 1/2)**
>
> We sincerely thank you for your positive assessment of our work, particularly for highlighting the principled nature of our approach and the strong generalization results. We appreciate your constructive feedback regarding ablation studies and stability analysis. Below, we address your specific comments point-by-point with additional data and analysis.
>
> ## 1. Lack of ablation studies
>
> We agree that dissecting the contributions of individual modules is crucial. We have added a comprehensive ablation study in **Table 7** (Appendix G) of the revised manuscript, isolating the effects of Positional Embeddings (PE), Channel Embeddings (CE), and the Mixture of Experts (MoE) strategy ($k=16$ vs. $k=1$).
>
> The results definitively validate the necessity of each component:
>
> - Channel Embeddings (CE): Removing CE results in catastrophic performance drops, specifically a **21.12% decrease in F1 score on ADFTD** and **13.57% on PTB-XL**. This confirms that while the backbone captures temporal dynamics, CE is essential for resolving the spatial topology inherent in multivariate medical data.
> - Positional Embeddings (PE): Removing PE causes a **17.40% F1 drop on ADFTD**, indicating that the permutation-invariant attention mechanism requires explicit temporal anchoring to decode sequential medical pathologies.
> - Mixture of Experts (MoE): Reducing the query capacity from $k=16$ to $k=1$ leads to a **16.88% drop on ADFTD**, demonstrating that a single prototype is insufficient to capture the complex, multi-modal distributions of disease classes.
>
> ## 2. Hyperparameter stability for channel embeddings and label queries
>
> We would like to clarify the parameterization of our embedding modules:
>
> 1. **Channel Embeddings:** There are **no tunable hyperparameters** for this module. The embedding dimension $D$ is strictly tied to the backbone model's hidden dimension, and the number of embeddings $C$ is determined automatically by the dataset's channel configuration. We can only turn on and off this module (shown in previous ablation studies).
> 2. **Label Queries:** The primary hyperparameter is the factor $k$ (number of queries per class). We have conducted an adapt-time scaling study presented in **Figure 5 and Table 5**. The results show that FORMED exhibits robust stability, with performance following a consistent power-law improvement as $k$ increases. The model performs reliably across a wide range of $k$ (from 64 to 2048), indicating that it is not sensitive to precise tuning but rather benefits predictably from increased capacity.
>
> ## 3. Stability in the few-shot regime
>
> We analyzed stability in the few-shot regime using the out-of-domain datasets (ECG200 and StandWalkJump), as detailed in **Table 5**. We observed that:
>
> - **Baseline Instability:** The baseline TimesFM-TSA (CNN head) exhibits high variance (large standard deviations) and often fails to converge effectively due to the scarcity of samples (e.g., only 27 total samples in SWJ, 200 in ECG200).
> - **FORMED Stability:** While FORMED with low $k$ also faces stability challenges, increasing the query capacity ($k \ge 512$) significantly stabilizes the model. For instance, on ECG200, increasing $k$ to 1024 not only improves Accuracy to **88.42%** but also reduces the standard deviation compared to lower $k$ settings. The MoE design acts as a robust, non-parametric density estimator, allowing the model to cover diverse class manifestations even with limited examples, thereby mitigating the brittleness typically seen in few-shot learning.

---

> ### Author Response · Authors · 2025-11-23
> **Response (part 2/2)**
>
> ## 4. Specificity to medical time series in model architecture
>
> The architecture of FORMED is explicitly engineered to address the fundamental challenges of **Medical Time Series (MedTS) heterogeneity**, specifically the varying number of channels, inconsistent sequence lengths, and diverse diagnostic tasks (binary vs. multi-class) found across datasets.
>
> Unlike standard time series models that often assume fixed input dimensions, our **Shared Decoding Attention (SDA)** and **Label Query (LQ)** mechanisms effectively disentangle domain-invariant knowledge from task-specific configurations. This design allows the model to:
>
> 1. Dynamically adapt to any channel configuration (via Channel Embeddings) without architectural changes.
> 2. Learn a "universal medical grammar" in the shared SDA that transfers across modalities (e.g., from EEG to ECG).
>
> While we focused on EEG and ECG, this rigorous separation of topology (Channel Embeddings) and semantics (Label Queries) makes the architecture theoretically applicable to other physiological signals like PPG or EMG, ensuring broad relevance to the medical domain. In theory, this should also be applicable to other time series domains should there be similar challenges or properties, which could be a valuable future work.
>
> ## 5. Applicability to other tabular foundation models
>
> We acknowledge this as a valuable direction for future research. FORMED is designed as a model-agnostic wrapper; its core logic, i.e., using learnable queries to decode frozen backbone features, is not strictly bound to TimesFM. It can theoretically be applied to any foundation model that outputs sequence representations. However, the current implementation relies on the backbone's ability to handle continuous time-series tokens. Extending this to tabular foundation models would require backbones that support similar tokenization strategies for irregular or categorical data. We have extended the discussion on this potential and its limitations in the Conclusion section.

---

### Author Response · Authors · 2025-11-23
**Response to Reviewers**

We sincerely thank all the reviewers for their time and insightful comments. We appreciate the positive assessment of our work’s novelty, particularly regarding the principled nature of the **FORMED** framework and its ability to address the heterogeneity of medical time series.

The reviewers raised important questions regarding baseline comparisons, component necessity, and experimental rigor. In response, we have conducted extensive new experiments and the results consistently support the effectiveness of FORMED.

## 1. Extended Baseline Comparisons (MiniROCKET, LoRA, PatchTST, etc.)

Addressing the feedback from **Reviewers VzRy and xKQU**, we have significantly expanded our baseline suite. We implemented **MiniROCKET** (a SOTA non-deep learning baseline), **LoRA** (Parameter-Efficient Fine-Tuning on TimesFM), a **PatchTST-based** variant of our framework, and **MLP-based** classifier for TimesFM. The results, now detailed in **Table 6**, demonstrate that FORMED consistently outperforms these methods. Specifically, we show that simple adaptations (like MLP heads or LoRA) are insufficient for the complex task in medical diagnostics, validating the indispensability of our Shared Decoding Attention architecture.

## 2. Comprehensive Ablation Studies

Following the suggestions of **Reviewers fXsE, VzRy, and hoQL**, we have added a detailed ablation study in **Table 7 (Appendix G)**. We systematically isolated the contributions of **Channel Embeddings (CE)**, **Positional Embeddings (PE)**, and the **Mixture of Experts (MoE)** strategy. The results confirm that removing any single component leads to catastrophic performance drops (e.g., 20% F1 drop without CE), proving that each module is essential for handling the spatiotemporal topology of multivariate medical data.

## 3. Stability Analysis in Few-Shot Regimes

In response to **Reviewers fXsE and hoQL**, we analyzed the model's behavior on small datasets (ECG200, SWJ). We found that while baselines exhibit high variance, FORMED stabilizes significantly as the query capacity ($k$) increases, following a predictable power law. These results are now discussed in **Table 5**.

## 4. Data Pipeline Audit and Correction

Prompted by **Reviewer VzRy**, we performed a thorough audit of our evaluation pipeline. We identified a bug in our result summarization script that affected the reporting for the TSA and GA baseline models. We have corrected this error throughout the manuscript (**Table 4, Figures 4, 7-11**). Importantly, even after correction, **FORMED maintains a significant performance lead** (e.g., +21.91% over MiniROCKET, +13.08% over TimesFM-TSA on ADFTD).

## Summary of Changes to the Manuscript

We have revised the manuscript to include the new results and changes mentioned above.
_(Note: In the revised manuscript, text in **red** indicates newly added information, and text in **blue** indicates changes to existing content.)_

- **(Added) Appendix F, Table 6:** New extended baseline comparisons (MiniROCKET, PatchTST + FORMED, TimesFM + MLP, TimesFM + MLP + LoRA, TimesFM + Attn).
- **(Added) Appendix G, Table 7:** New ablation studies on components in FORMED (w/o PE, w/o CE, w/o PE & CE, w/o MoE).
- **(Updated) Table 4, Figure 4, Figure 7-11:** Updated results for TSA and GA models after fixing the summarization script bug.
- **(Updated) Abstract:** Included an anonymized GitHub repository link to ensure full reproducibility.
- **(Enhanced) Section 1 Introduction:** Clarified the attribution of the MedTS cohort curation to prior work (Medformer).
- **(Enhanced) Section 2 Related Work:** Clarified the landscape of time series foundation models, noting the lack of models designed exclusively for classification, and added citations on related adaptation techniques (e.g., Gen-P-Tuning).
- **(Enhanced) Section 3 Problem Statement:** Explicitly defined "medical time series" as continuous waveform data (e.g., EEG, ECG) within the scope of this work, distinguishing it from sparse EHR data.
- **(Enhanced) Subsection 4.2 Classifier Design:** Clarified the function and parameterization of Channel Embeddings.
- **(Enhanced) Subsection 5.1 Evaluation on Repurposing:** Integrated the extended baseline comparison (Table 6) to justify the architectural complexity of FORMED over simple output modifications.
- **(Enhanced) Subsection 5.2 Evaluation on Adapting:** Provided deeper analysis on the stability of FORMED in few-shot regimes compared to baselines.
- **(Enhanced) Subsection 6 Discussion and Conclusion:** Extended the discussion to cover applicability to other modalities (low-frequency signals) and other backbone models.

---

### Author Response · Authors · 2025-11-29

Dear Area Chair,

We understand that due to the recent OpenReview security incident and the subsequent administrative reset, the discussion phase has come to a _de facto_ end, and the updated score trajectory (improving from **8 / 6 / 4 / 4** to **8 / 6 / 6 / 6**) is no longer visible to you.

We provide the review timeline below, indicating the score increasing was established through constructive rebuttal and discussion prior to the administrative reset:
- Reviewer xKQU raised the score to 6 on November 23rd, **before the indicdent,**
- Reviewer VzRy raised the score to 6 on November 27th, **on the day of the incident**.


Moreover, to assist your assessment, we provide this summary of the dialogue that occurred during the rebuttal period and highlight where the resolving evidence can be found in our revised manuscript.

## 1. Consensus on Strengths

The reviewers have reached a consensus on the novelty and significance of the work:

- **Principled Architecture**: Reviewers xKQU and fXsE praised the **"clean and elegant" strategy** of separating shared domain knowledge from task-specific knowledge, noting **the approach is "principled and well justified"**.
- **Critical Problem Solving**: Reviewer hoQL highlighted the work’s value in **addressing "inter- and intra-dataset heterogeneity,"** a major barrier in medical AI.
- **Strong Generalization**: The reviewers recognized the model's ability to **generalize to unseen patients and tasks** as a key contribution.

## 2. Resolution of All Weaknesses

The initial weaknesses and questions focused on baselines and ablation. These were addressed through extensive new experiments added during the rebuttal, which drove the score improvements. For a more detailed summary of changes, please refer to [Response to Reviewers](https://openreview.net/forum?id=wNEzRYiyZM&noteId=e0DhGO5Pvc).

- **Strength of Baselines (Reviewers VzRy, xKQU)**

  - _Comments_: Comparisons against strong classifiers (_e.g._, **MiniRocket**) and Parameter-Efficient Fine-Tuning methods (**LoRA**) to prove the architecture's necessity
  - _Response_: We added **Table 6 (Appendix F)**. The results show FORMED (63.66% F1 on ADFTD) significantly outperforms **MiniRocket** (45.41%) and **TimesFM+LoRA** (49.23%). This empirically proves that simple fine-tuning or fixed kernels are insufficient for medical heterogeneity and validates our "Repurposing" approach.

- **Ablation and Necessity of Components (Reviewers fXsE, hoQL, VzRy)**

  - _Comments_: Justification of the necessity for the design components, including Channel Embedding (CE), Position Embedding (PE), and Mixture-of-Experts (MoE).
  - _Response_: We added **Table 7 (Appendix G)**. The study demonstrates that removing any module causes a catastrophic performance drop (_e.g._, -21% F1 on ADFTD w/o CE), proving that our designs are well-motivated and functional.

- **Stability in Few-Shot Regimes (Reviewer fXsE, hoQL)**

  - _Comments_: Demonstration of hyperparameter stability and performance on small datasets.
  - _Response_: We added discussion in **Table 5**, which highlights that FORMED's performance scales robustly with the number of query vectors ($k$), achieving high stability (low variance) even on the small SWJ dataset with only 27 samples in total.

## 3. Conclusion

Our rebuttal successfully addressed the reviewers' questions, transforming the paper into a robust contribution backed by strong empirical evidence. We have updated the manuscript to contain these crucial new results and analysis.

**We sincerely thank Reviewers xKQU, hoQL, VzRy, and fXsE for their high-quality feedback which substantially improved the paper. We also extend our gratitude to you for taking on this assignment under these exceptional circumstances.**

Sincerely,
The Authors of #2880

---

### Meta-Review · Area_Chair_NVth · 2026-01-07

**Summary:**

reviewers gave scores of 4,4,6,8. the main concerns include:

Experimental baselines: the baselines being compared to are mostly models intended for time series forecasting, not classification. Even within the forecasting domain, SOTA models are not represented, and some of the included baselines.

Existing parameter-efficient fine tuning methods like LoRA (or even end-to-end fine-tuning) could be used for either the Repurposing or Adapting phase rather than the new method the authors propose, but these are not discussed or evaluated against.

Ablations showing the contributions of different modeling decisions are not provided.

Background information provided is inaccurate and missing references.

Novelty in light of "Generalized Prompt Tuning: Adapting Frozen Univariate Time Series Foundation Models for Multivariate Healthcare Time Series".

Need experiments of how FORMED works when using different base time series foundation models.

Need other kinds of medical time series data such as EHR data (e.g., there are ways to preprocess MIMIC for instance to get it to be regularly sampled and to consider classification tasks).

**Reviewer Concerns:**

Experimental baselines: the baselines being compared to are mostly models intended for time series forecasting, not classification. Even within the forecasting domain, SOTA models are not represented, and some of the included baselines.

--> authors added comparisons with MiniROCKET and showed improvements.

Existing parameter-efficient fine tuning methods like LoRA (or even end-to-end fine-tuning) could be used for either the Repurposing or Adapting phase rather than the new method the authors propose, but these are not discussed or evaluated against.

--> authors added comparison with PEFT (LoRA) and Full Fine-Tuning and showed improvements.

Ablations showing the contributions of different modeling decisions are not provided.

--> authors added

Background information provided is inaccurate and missing references.

--> addressed

Novelty in light of "Generalized Prompt Tuning: Adapting Frozen Univariate Time Series Foundation Models for Multivariate Healthcare Time Series".

--> authors discussed some differences but did not make direct empirical comparisons, concern still outstanding

Need experiments of how FORMED works when using different base time series foundation models.

--> did not add, concern still outstanding

Need other kinds of medical time series data such as EHR data (e.g., there are ways to preprocess MIMIC for instance to get it to be regularly sampled and to consider classification tasks).

--> addressed

**Reviewer Scores:**

i think the 2 reviewers who gave 4 have a chance of increasing to 6, most of their comments were addressed.

---

### Decision · Program_Chairs · 2026-01-26

Accept (Poster)